# Lower Bounds and Optimal Algorithms for Personalized Federated Learning

**Filip Hanzely**
KAUST
Thuwal, Saudi Arabia
filip.hanzely@kaust.edu.sa

**Slavomír Hanzely**
KAUST
Thuwal, Saudi Arabia
slavomir.hanzely@kaust.edu.sa

**Samuel Horváth**
KAUST
Thuwal, Saudi Arabia
samuel.horvath@kaust.edu.sa

**Peter Richtárik**
KAUST
Thuwal, Saudi Arabia
peter.richtarik@kaust.edu.sa

## Abstract

In this work, we consider the optimization formulation of personalized federated learning recently introduced by [19] which was shown to give an alternative explanation to the workings of local `SGD` methods. Our first contribution is establishing the first lower bounds for this formulation, for both the communication complexity and the local oracle complexity. Our second contribution is the design of several optimal methods matching these lower bounds in almost all regimes. These are the first provably optimal methods for personalized federated learning. Our optimal methods include an accelerated variant of `FedProx`, and an accelerated variance-reduced version of `FedAvg`/Local SGD. We demonstrate the practical superiority of our methods through extensive numerical experiments.

## 1 Introduction

Federated Learning (FL) [32, 24] is a relatively new field attracting much attention lately. Specifically, FL is a subfield of distributed machine learning that aims to fit the data stored locally on a very large number of heterogeneous clients. Unlike typical distributed learning inside a data center, each FL client only sees his/her data, which might differ from the population average significantly. Furthermore, as the clients are often physically located far away from the central server, communication becomes a notable bottleneck, which is far more significant compared to centralized datacenter learning.

While the key factor differentiating FL from other varieties of supervised learning lies in the practical considerations coming from the need to train models from private and heterogeneous data stored across a large number of heterogeneous and not always available devices, FL is supervised learning, and as such shares the same mathematical formalism. In particular, the standard formulation of FL aims to find the minimizer of the overall population loss

$$\min_{z \in \mathbb{R}^d} \frac{1}{n} \sum_{i=1}^n f_i(z) = \min_{x_1, x_2, \ldots, x_n \in \mathbb{R}^d} \frac{1}{n} \sum_{i=1}^n f_i(x_i), \tag{1}$$
$$\text{subject to } x_1 = x_2 = \cdots = x_n,$$

where $f_i$ is the loss of the client $i$ that only depends on his/her own local data.

However, the requirements of several FL applications [53, 25, 10] question the appropriateness of the objective (1) as a modeling paradigm for FL. Specifically, the minimizer of the overall population

loss might not be the ideal model for a given client given that his/her data distribution differs from the population significantly. A good example to illustrate the requirement of *personalized* FL models is next word prediction on mobile devices, where a personalized FL approach [20] significantly outperforms the standard non-personalized one.

There are multiple strategies in the literature for incorporating e personalization into FL: multi-task learning [49, 48, 14], transfer learning [54, 23], variational inference [8], mixing of local and global models [41, 19, 31, 10] and others [13]. See also [25, 22] for a survey on personalized FL.

In this work, we focus on the mixing FL objective from [19], which is well-known from the area of distributed optimization [26, 16] and distributed transfer learning [30, 50]. The mentioned formulation allows the local models $x_i$ to be mutually different, while penalizing their dissimilarity:

$$
\min_{x=[x_1,\ldots,x_n]\in\mathbb{R}^{nd},\forall i:\, x_i\in\mathbb{R}^d} \left\{ F(x) := \underbrace{\frac{1}{n}\sum_{i=1}^n f_i(x_i)}_{:=f(x)} + \lambda \underbrace{\frac{1}{2n}\sum_{i=1}^n \|x_i - \bar{x}\|^2}_{:=\psi(x)} \right\}. \tag{2}
$$

Surprisingly, the optimal solution $x^\star = [x_1^\star, x_2^\star, \ldots, x_n^\star] \in \mathbb{R}^{nd}$ of (2) can be expressed as $x_i^\star = \bar{x}^\star - \frac{1}{\lambda}\nabla f_i(x_i^\star)$, where $\bar{x}^\star = \frac{1}{n}\sum_{i=1}^n x_i^\star$ [19], which strongly resembles the influential model-agnostic meta learning (MAML) framework from [15].

Specifically, it was shown that a simple version of Stochastic Gradient Descent (SGD) applied on (2) is essentially[1] equivalent to FedAvg algorithm [19]. Furthermore, the FL formulation (2) enabled local gradient methods to outperform their non-local cousins when applied to heterogeneous data problems.[2]

## 2 Contributions

In this paper, we study the personalized FL formulation (2). We propose lower complexity bounds for communication and local computation, and develop several algorithms capable of achieving them. Our contributions can be summarized as follows:

- We propose a *lower bound on the communication complexity* of the federated learning formulation (2). We show that for any algorithm that satisfies a certain reasonable assumption (see As. 3.1) there is an instance of (2) with $L$-smooth, $\mu$-strongly convex[3] local objectives $f_i$ requiring at least $\mathcal{O}\left(\sqrt{\frac{\min\{L,\lambda\}}{\mu}}\log\frac{1}{\varepsilon}\right)$ communication rounds to get to the $\varepsilon$-neighborhood of the optimum.

- We further establish a *lower complexity bound on the number of local oracle calls*. We show that one requires at least $\mathcal{O}\left(\sqrt{\frac{\min\{L,\lambda\}}{\mu}}\log\frac{1}{\varepsilon}\right)$ proximal oracle calls[4] or at least $\mathcal{O}\left(\sqrt{\frac{L}{\mu}}\log\frac{1}{\varepsilon}\right)$ evaluations of local gradients. Similarly, given that each of the local objectives is of a $m$-finite-sum structure with $\tilde{L}$-smooth summands, we show that at least $\mathcal{O}\left(\left(m+\sqrt{\frac{m\tilde{L}}{\mu}}\right)\log\frac{1}{\varepsilon}\right)$ gradients of the local summands are required.

| Algorithm | Local oracle | Optimal # comm | Optimal # local |
|---|---|---|---|
| L2GD [19] | Grad | ✗ | ✗ |
| L2SGD+ [19] | Stoch grad | ✗ | ✗ |
| APGD1 [50] (A. 2) | Prox | ✓ (if $\lambda \leq L$) | ✓ (if $\lambda \leq L$) |
| APGD2 [50] (A. 3) | Grad | ✓ (if $\lambda \geq L$) | ✓ |
| APGD2 [50] (A. 3) | Stoch grad | ✓ (if $\lambda \geq L$) | ✗ |
| IAPGD [50] (A. 1) + AGD [38] | Grad | ✓ (if $\lambda \leq L$) | ✓ (if $\lambda \leq L$) |
| IAPGD [50] (A. 1) + Katyusha [3] | Stoch grad | ✓ (if $\lambda \leq L$) | ✓ (if $m\lambda \leq \tilde{L}$) |
| AL2SGD+ (A. 4) | Stoch grad | ✓ | ✓ $\left(\text{if } \lambda \leq \tilde{L}\right)$ |

Table 1: Algorithms for solving (2) and their (optimal) complexities.

- We develop several approaches for solving (2) achieving *optimal communication complexity and optimal local gradient complexity* under various circumstances. Specializing the approach from [50] to our problem, we apply Accelerated Proximal Gradient Descent (APGD) in two different ways – we either take a gradient step with respect to $f$ and a proximal step with respect to $\lambda\psi$, or vice versa. In the first case, we get communication complexity and local gradient complexity of the order $\mathcal{O}\left(\sqrt{\frac{L}{\mu}}\log\frac{1}{\varepsilon}\right)$, which is optimal if $L \leq \lambda$. In the second case, we get communication complexity and local prox complexity of the order $\mathcal{O}\left(\sqrt{\frac{\lambda}{\mu}}\log\frac{1}{\varepsilon}\right)$, which is optimal if $L \geq \lambda$. Motivated again by [50], we argue that local prox steps can be evaluated inexactly[5] either by running Accelerated Gradient Descent (AGD) [38] locally, or by running Katyusha [3] locally, given that the local objective is of a $m$-finite sum structure with $\tilde{L}$-smooth summands. Local AGD approach preserves $\mathcal{O}\left(\sqrt{\frac{\lambda}{\mu}}\log\frac{1}{\varepsilon}\right)$ communication complexity and yields $\tilde{\mathcal{O}}\left(\sqrt{\frac{L+\lambda}{\mu}}\right)$ local gradient complexity, both of them optimal for $L \geq \lambda$ (up to log factors). Similarly, employing Katyusha locally, we obtain communication complexity of order $\mathcal{O}\left(\sqrt{\frac{\lambda}{\mu}}\log\frac{1}{\varepsilon}\right)$ and local gradient complexity of order $\tilde{\mathcal{O}}\left(m\sqrt{\frac{\lambda}{\mu}} + \sqrt{\frac{m\tilde{L}}{\mu}}\right)$. The former is optimal once $L \geq \lambda$, while the latter is (up to log factor) optimal once $m\lambda \leq \tilde{L}$.

- Inexact APGD with local randomized solver has three drawbacks: (i) there are extra log factors in the local gradient complexity, (ii) boundedness of the algorithm iterates as an assumption is required, and (iii) the communication complexity is suboptimal for $\lambda > L$. In order to fix all the issues, *we accelerate the* L2SGD+ *algorithm* from [19]. The proposed algorithm, AL2SGD+, enjoys the optimal communication complexity $\mathcal{O}\left(\sqrt{\frac{\min\{\tilde{L},\lambda\}}{\mu}}\log\frac{1}{\varepsilon}\right)$ and the local summand gradient complexity $\mathcal{O}\left(\left(m + \sqrt{\frac{m(\tilde{L}+\lambda)}{\mu}}\right)\log\frac{1}{\varepsilon}\right)$, which is optimal for $\lambda \leq \tilde{L}$. Unfortunately, the two bounds are not achieved at the same time, as we shall see.

- As a consequence of all aforementioned points, **we show the optimality of local algorithms applied on FL problem** (2) **with heterogeneous data**. We believe this is an important contributions to the FL literature. Until now, local algorithms were known to be optimal only when all nodes own the same data set, which is not a realistic regime for FL applications. By showing the optimality of local methods, we partially justify standard FL practices (i.e., using local methods in the practical scenarios with non-iid data), albeit with a twist: local methods should be seen as methods for solvimg the personalized FL formulation (2) first introduced in [19].

Table 1 presents a summary of the described results: for each algorithm, it indicates the local oracle requirement and the circumstances under which the corresponding complexities are optimal.

| Local oracle | Optimal # Comm | Optimal # Local calls | Algorithm |
|---|---|---|---|
| Proximal | ✓ | ✓ | $\begin{cases} \lambda \geq L: & \texttt{APGD2}\,[50](\text{A. }3) \\ \lambda \leq L: & \texttt{APGD1}\,[50](\text{A. }2) \end{cases}$ |
| Gradient | ✓ | ✓ | $\begin{cases} \lambda \geq L: & \texttt{APGD2}\,[50](\text{A. }3) \\ \lambda \leq L: & \texttt{IAPGD}\,[50](\text{A. }1) + \texttt{AGD}\,[38] \end{cases}$ |
| Stoch grad | ✓ | ✓ if $m\lambda \leq \tilde{L}$ | $\begin{cases} \lambda \geq L: & \texttt{APGD2}\,[50](\text{A. }3) \\ \lambda \leq L: & \texttt{IAPGD}\,[50](\text{A. }1) + \texttt{Katyusha}\,[3] \end{cases}$ |
| Stoch grad | ✓ ✗ | ✗ ✓ if $\lambda \leq \tilde{L}$ | $\texttt{AL2SGD+}^{(*)}$ |

Table 2: Matching (up to $\log$ and constant factors) lower and upper complexity bounds for solving (2). Indicator ✓ means that the lower and upper bound are matching up to constant and log factors, while ✗ means the opposite. $^{(*)}$ (`AL2SGD+` under stochastic gradient oracle): `AL2SGD+` can be optimal either in terms of the communication or in terms of the local computation; the two cases require a slightly different parameter setup.

**Optimality.** Next we present Table 2 which carries an information orthogonal to Table 1. In particular, Table 2 indicates whether our lower and upper complexities match for a given pair of {local oracle, type of complexity}. The lower and upper complexity bounds on the number of communication rounds match regardless of the local oracle. Similarly, the local oracle calls match almost always with one exception when the local oracle provides summand gradients and $\lambda > \tilde{L}$.

**Remark 2.1** *Our upper and lower bounds do not match for the local summand gradient oracle once we are in the classical FL setup* (2), *which we recover for $\lambda = \infty$. In such a case, an optimal algorithm was developed only very recently [21] under a slightly stronger oracle – the proximal oracle for the local summands.*

## 3 Lower complexity bounds

Before stating our lower complexity bounds for solving (2), let us formalize the notion of an oracle that an algorithm interacts with.

As we are interested in both communication and local computation, we will also distinguish between two different oracles: the *communication oracle* and the *local oracle*. While the communication oracle allows the optimization history to be shared among the clients, the local oracle $\text{Loc}(x_i, i)$ provides either a local proximal operator, local gradient, or local gradient of a summand given that a local loss is of a finite-sum structure itself ($f_i(x_i) = \frac{1}{m} \sum_{j=1}^{m} \tilde{f}_{i,j}(x_i)$):

$$\text{Loc}(x, i) = \begin{cases} \{\nabla f_i(x_i), \text{prox}_{\beta_i f_i}(x_i)\} & \text{if oracle is } \textit{proximal} \text{ (for any } \beta_i \geq 0) \\ \{\nabla f_i(x_i)\} & \text{if oracle is } \textit{gradient} \\ \{\nabla \tilde{f}_{i,j_i}(x_i)\} & \text{if oracle is } \textit{summand gradient} \text{ (for any } 1 \leq j_i \leq m) \end{cases}$$

for all clients $i$ simultaneously, which we refer to as a single local oracle call.

Next, we restrict ourselves to algorithms whose iterates lie in the span of previously observed oracle queries. Assumption 3.1 formalizes this.

**Assumption 3.1** *Let $\{x^k\}_{k=1}^{\infty}$ be iterates generated by algorithm $\mathcal{A}$. For $1 \leq i \leq n$ let $\{S_i^k\}_{k=0}^{\infty}$ be a sequence of sets defined recursively as follows:*

$$S_i^0 = \text{Span}(x_i^0)$$

$$S_i^{k+1} = \begin{cases} \text{Span}\left(S_i^k, \text{Loc}(x^k, i)\right) & \textit{if } \zeta(k) = 1 \\ \text{Span}\left(S_1^k, S_2^k, \ldots, S_n^k\right) & \textit{otherwise,} \end{cases}$$

*where $\zeta(k) = 1$ if the local oracle was queried at the iteration $k$, otherwise $\zeta(k) = 0$. Then, assume that $x_i^k \in S_i^k$.*

Assumption 3.1 is rahter standard in the literature of distributed optimization [44, 21]; it informally means that the iterates of $\mathcal{A}$ lie in the span of explored directions only. A similar restriction is in place for several standard optimization lower complexity bounds [37, 27]. We shall, however, note that Assumption 3.1 can be omitted by choosing the worst-case objective adversarially based on the algorithm decisions [36, 51, 52]. We do not explore this direction for the sake of simplicity.

## 3.1 Lower complexity bounds on the communication

We now present our lower bound on the communication complexity of problem (2).

**Theorem 3.1** *Let $k \geq 0, L \geq \mu, \lambda \geq \mu$. Then, there exist $L$-smooth $\mu$-strongly convex functions $f_1, f_2, \ldots f_n : \mathbb{R}^d \to \mathbb{R}$ and a starting point $x^0 \in \mathbb{R}^{nd}$, such that the sequence of iterates $\{x^t\}_{t=1}^{k}$ generated by any algorithm $\mathcal{A}$ meeting Assumption 3.1 satisfies*

$$\|x^k - x^\star\|^2 \geq \frac{1}{4}\left(1 - 10\max\left\{\sqrt{\frac{\mu}{\lambda}}, \sqrt{\frac{\mu}{L-\mu}}\right\}\right)^{C(k)+1}\|x^0 - x^\star\|^2. \tag{3}$$

*Above, $C(k)$ stands for the number of communication oracle queries at the first $k$ iterations of $\mathcal{A}$.*

Theorem 3.1 shows that in order get to $\varepsilon$ close to the optimum, one needs at least $\mathcal{O}\left(\sqrt{\frac{\min\{L,\lambda\}}{\mu}}\log\frac{1}{\varepsilon}\right)$ rounds of the communications. This reduces to known communication complexity $\mathcal{O}\left(\sqrt{\frac{L}{\mu}}\log\frac{1}{\varepsilon}\right)$ for standard FL objective (1) from [44, 21] when $\lambda = \infty$.[6]

## 3.2 Lower complexity bounds on the local computation

Next, we present our lower complexity bounds on the number of the local oracle calls for three different types of a local oracle. In a special case when $\lambda = \infty$, we recover known local oracle bounds for the classical FL objective (1) from [21].

**Proximal oracle.** The construction from Theorem 3.1 not only requires $\mathcal{O}\left(\sqrt{\frac{\min\{\lambda,L\}}{\mu}}\log\frac{1}{\varepsilon}\right)$ communication rounds to reach $\varepsilon$-neighborhood of the optimum, it also requires at least $\mathcal{O}\left(\sqrt{\frac{\min\{\lambda,L\}}{\mu}}\log\frac{1}{\varepsilon}\right)$ calls of any local oracle, which serves as the lower bound on the local proximal oracle.

**Gradient oracle.** Setting $x^0 = 0 \in \mathbb{R}^{nd}$ and $f_1 = f_2 = \cdots = f_n$, problem (2) reduces to minimizing a single local objective $f_1$. Selecting next $f_1$ as the worst-case quadratic function from [37], the corresponding objective requires at least $\mathcal{O}\left(\sqrt{\frac{L}{\mu}}\log\frac{1}{\varepsilon}\right)$ gradient calls to reach $\varepsilon$-neighborhood, which serves as our lower bound. Note that parallelism does not help as the starting point is identical on all machines and the construction of $f$ only allows to explore a single coordinate per a local call, regardless of communication.

**Summand gradient oracle.** Suppose that $\tilde{f}_{i,j}$ is $\tilde{L}$-smooth for all $1 \leq j \leq m$ and $1 \leq i \leq n$. Let us restrict attention on the class of client-symmetric algorithms for which $x_i^{k+1} = \mathcal{A}(H_i^k, H_{-i}^k, C^k)$, where $H_i$ is history of local gradients gathered by client $i$, $H_{-i}$ is an *unordered* set with elements $H_l$ for all $l \neq i$ and $C^k$ are indices of the communication rounds of the past. We assume that $\mathcal{A}$ is either deterministic, or generated from given seed that is identical for all clients initially.[7] Setting again $x^0 = 0 \in \mathbb{R}^{nd}$ and $f_1 = f_2 = \cdots = f_n$, the described algorithm restriction yields $x_1^k = x_2^k = \cdots = x_n^k$ for all $k \geq 0$. Consequently, the problem reduces to minimize a single finite sum objective $f_1$ which requires at least $\mathcal{O}\left(m + \sqrt{\frac{m\tilde{L}}{\mu}}\log\frac{1}{\varepsilon}\right)$ summand gradient calls [27, 51].

# 4 Optimal algorithms

In this section, we present several algorithms that match the lower complexity bound on the number of communication rounds and the local steps obtained in Section 3.

## 4.1 Accelerated Proximal Gradient Descent (APGD) for Federated Learning

The first algorithm we mention is a version of the accelerated proximal gradient descent [6]. In order to see how the method specializes in our setup, let us first describe the non-accelerated counterpart – proximal gradient descent (PGD).

Let a function $h : \mathbb{R}^{nd} \to \mathbb{R}$ be $L_h$-smooth and $\mu_h$-strongly convex, and function $\phi : \mathbb{R}^{nd} \to \mathbb{R} \cup \{\infty\}$ be convex. In its most basic form, iterates of PGD to minimize a regularized convex objective $h(x) + \phi(x)$ are generated recursively as follows

$$x^{k+1} = \text{prox}_{\frac{1}{L_h}\phi}\left(x^k - \frac{1}{L_h}\nabla h(x^k)\right) = \underset{x \in \mathbb{R}^{nd}}{\text{argmin }} \phi(x) - \frac{L_h}{2}\left\|x - \left(x^k - \frac{1}{L_h}\nabla h(x^k)\right)\right\|^2. \quad (4)$$

The iteration complexity of the above process is $\mathcal{O}\left(\frac{L_h}{\mu_h}\log\frac{1}{\varepsilon}\right)$.

Motivated by [50][8], there are two different ways to apply the process (4) to the problem (2). A more straightforward option is to set $h = f, \phi = \lambda\psi$, which results in the following update rule

$$x_i^{k+1} = \frac{Ly_i^k + \lambda\bar{y}^k}{L + \lambda}, \quad \text{where} \quad y_i^k = x_i^k - \frac{1}{L}\nabla f(x_i^k), \quad \bar{y}^k = \frac{1}{n}\sum_{i=1}^{n} y_i^k, \quad (5)$$

and it yields $\mathcal{O}\left(\frac{L}{\mu}\log\frac{1}{\varepsilon}\right)$ rate. The second option is to set $h(x) = \lambda\psi(x) + \frac{\mu}{2n}\|x\|^2$ and $\phi(x) = f(x) - \frac{\mu}{2n}\|x\|^2$. Consequently, the update rule (4) becomes (see Lemma C.3 in the Appendix):

$$x_i^{k+1} = \text{prox}_{\frac{1}{\lambda}f_i}(\bar{x}^k) = \underset{z \in \mathbb{R}^d}{\text{argmin }} f_i(z) + \frac{\lambda}{2}\|z - \bar{x}^k\|^2 \quad \text{for all } i, \quad (6)$$

matching the FedProx [28] algorithm. The iteration complexity we obtain is, however, $\mathcal{O}\left(\frac{\lambda}{\mu}\log\frac{1}{\varepsilon}\right)$ (see Lemma C.3 again).

As both (5) and (6) require a single communication round per iteration, the corresponding communication complexity becomes $\mathcal{O}\left(\frac{L}{\mu}\log\frac{1}{\varepsilon}\right)$ and $\mathcal{O}\left(\frac{\lambda}{\mu}\log\frac{1}{\varepsilon}\right)$ respectively, which is suboptimal in the light of Theorem 3.1.

Fortunately, incorporating the Nesterov's momentum [38, 6] on top of the procedure (6) yields both an optimal communication complexity and optimal local prox complexity once $\lambda \leq L$. We will refer to such method as APGD1 (Algorithm 2 in the Appendix). Similarly, incorporating the acceleration into (5) yields both an optimal communication complexity and optimal local prox complexity once $\lambda \geq L$. Furthermore, such an approach yields the optimal local gradient complexity regardless of the relative comparison of $L, \lambda$. We refer to such method APGD2 (Algorithm 3 in the Appendix).

## 4.2 Beyond proximal oracle: Inexact APGD (IAPGD)

In most cases, the local proximal oracle is impractical as it requires the exact minimization of the regularized local problem at each iteration. In this section, we describe an accelerated inexact [45] version of (6) (Algorithm 1), which only requires a local (either full or summand) gradient oracle. We present two different approaches to achieve so: AGD [38] (under the gradient oracle) and Katyusha [3] (under the summand gradient oracle). Both strategies, however, share a common characteristic: they progressively increase the effort to inexactly evaluate the local prox, which is essential in order to preserve the optimal communication complexity.

**Algorithm 1** `IAPGD` + $\mathcal{A}$

---

**Require:** Starting point $y^0 = x^0 \in \mathbb{R}^{nd}$
  **for** $k = 0, 1, 2, \ldots$ **do**

    <span style="color:blue">Central server</span> computes the average $\bar{y}^k = \frac{1}{n} \sum\limits_{i=1}^{n} y_i^k$

    For all <span style="color:red">clients</span> $i = 1, \ldots, n$:
      Set $h_i^{k+1}(z) := f_i(z) + \frac{\lambda}{2}\|z - \bar{y}^k\|^2$ and find $x_i^{k+1}$ using local solver $\mathcal{A}$ for $T_k$ iterations

$$h_i^{k+1}(x_i^{k+1}) \leq \epsilon_k + \min_{z \in \mathbb{R}^d} h_i^{k+1}(z). \tag{7}$$

    For all <span style="color:red">clients</span> $i = 1, \ldots, n$: Take the momentum step $y_i^{k+1} = x_i^{k+1} + \frac{\sqrt{\lambda} - \sqrt{\mu}}{\sqrt{\lambda} + \sqrt{\mu}}(x_i^{k+1} - x_i^k)$

  **end for**

---

**Remark 4.1** *As already mentioned, the idea of applying* `IAPGD` *to solve* (2) *is not new; it was already explored in [50].[9] However, [50] does not argue about the optimality of* `IAPGD`. *Less importantly, our analysis is slightly more careful, and it supports* `Katyusha` *as a local sub-solver as well.*

`IAPGD` + `AGD`

The next theorem states the convergence rate of `IAPGD` with `AGD` [38] as a local subsolver.

**Theorem 4.2** *Suppose that $f_i$ is $L$-smooth and $\mu$-strongly convex for all $i$. Let* `AGD` *with starting point $y_i^k$ be employed for $T_k := \sqrt{\frac{L+\lambda}{\mu+\lambda}} \log\left(1152L\lambda n^2 \left(2\sqrt{\frac{\lambda}{\mu}} + 1\right)^2 \mu^{-2}\right) + 4\sqrt{\frac{\mu(L+\lambda)}{\lambda(\mu+\lambda)}} k$ iterations to approximately solve* (7) *at iteration $k$. Then, we have*

$$F(x^k) - F^\star \leq 8\left(1 - \sqrt{\frac{\mu}{\lambda}}\right)^k (F(x^0) - F^\star),$$

*where $F^\star = F(x^\star)$. As a result, the total number of communications required to reach $\varepsilon$-approximate solution is*

$$\mathcal{O}\left(\sqrt{\frac{\lambda}{\mu}} \log\frac{1}{\varepsilon}\right).$$

*The corresponding local gradient complexity is*

$$\mathcal{O}\left(\sqrt{\frac{L+\lambda}{\mu}} \log\frac{1}{\varepsilon}\left(\log\frac{L\lambda n}{\mu} + \log\frac{1}{\varepsilon}\right)\right) = \tilde{\mathcal{O}}\left(\sqrt{\frac{L+\lambda}{\mu}}\right).$$

As expected, the communication complexity of `IAPGD` + `AGD` is $\mathcal{O}\left(\sqrt{\frac{\lambda}{\mu}} \log\frac{1}{\varepsilon}\right)$, thus optimal. On the other hand, the local gradient complexity is $\tilde{\mathcal{O}}\left(\sqrt{\frac{L+\lambda}{\mu}}\right)$. For $\lambda = \mathcal{O}(L)$ this simplifies to $\tilde{\mathcal{O}}\left(\sqrt{\frac{L}{\mu}}\right)$, which is, up to $\log$ and constant factors identical to the lower bound on the local gradient calls.

`IAPGD` + `Katyusha`

In practice, the local objectives $f_i$'s often correspond to a loss of some model on the given client's data. In such a case, each function $f_i$ is of the finite-sum structure: $f_i(x_i) = \frac{1}{m} \sum_{j=1}^{m} \tilde{f}_{i,j}(x_i)$.

Clearly, if $m$ is large, solving the local subproblem with `AGD` is rather inefficient as it does not take an advantage of the finite-sum structure. To tackle this issue, we propose solving the local subproblem (7) using `Katyusha`.[10]

**Theorem 4.3** *Let $\tilde{f}_{i,j}$ be $\tilde{L}$-smooth and $f_i$ be $\mu$-strongly convex for all $1 \leq i \leq n, 1 \leq j \leq m$.[11]*

*Let Katyusha with starting point $y_i^k$ be employed for $T_k = \mathcal{O}\left(\left(m + \sqrt{m\frac{\tilde{L}+\lambda}{\mu+\lambda}}\right)\left(\log\frac{1}{R^2} + k\sqrt{\frac{\mu}{\lambda}}\right)\right)$ iterations to approximately solve* (7) *at iteration $k$ of* `IAPGD` *for some small $R$ (see proof for details). Given that the iterate sequence $\{x^k\}_{k=0}^{\infty}$ is bounded, the expected communication complexity of* `IAPGD+Katyusha` *is*

$$\mathcal{O}\left(\sqrt{\frac{\lambda}{\mu}}\log\frac{1}{\epsilon}\right),$$

*while the local summand gradient complexity is*

$$\tilde{\mathcal{O}}\left(m\sqrt{\frac{\lambda}{\mu}} + \sqrt{m\frac{\tilde{L}}{\mu}}\right).$$

Theorem 4.3 shows that local `Katyusha` enjoys the optimal communication complexity. Furthermore, if $\sqrt{m}\lambda = \mathcal{O}(\tilde{L})$, the total expected number of local gradients becomes optimal as well (see Sec. 3.2).

There is, however, a notable drawback of Theorem 4.3 over Theorem 4.2 – Theorem 4.3 requires a boundedness of the sequence $\{x^k\}_{k=0}^{\infty}$ as an assumption, while this piece is not required for `IAPGD+AGD` due to its deterministic nature. In the next section, we devise a stochastic algorithm `AL2SGD+` that does not require such an assumption. Furthermore, the local (summand) gradient complexity of `AL2SGD+` does not depend on the extra log factors, and, at the same time, `AL2SGD+` is optimal in a broader range of scenarios.

### 4.3   Accelerated `L2SGD+`

In this section, we introduce accelerated version of `L2SGD+` [19], which can be viewed as a variance-reduced variant of `FedAvg` devised to solve (2). The proposed algorithm, `AL2SGD+`, is stated as Algorithm 4 in the Appendix. From high-level point of view, `AL2SGD+` is nothing but `L-Katyusha` with non-uniform minibatch sampling.[12] In contrast to the approach from Section 4.2, `AL2SGD+` does not treat $f$ as a proximable regularizer, but rather directly constructs $g^k$–a non-uniform minibatch variance reduced stochastic estimator of $\nabla F(x^k)$. Next, we state the communication and the local summand gradient complexity of `AL2SGD+`.

**Theorem 4.4** *Suppose that the parameters of* `AL2SGD+` *are chosen as stated in Proposition E.4 in the Appendix. In such case, the communication complexity of* `AL2SGD+` *with $\rho = p(1-p)$, where $p = \frac{\lambda}{\lambda+\tilde{L}}$, is*

$$\mathcal{O}\left(\sqrt{\frac{\min\{\tilde{L},\lambda\}}{\mu}}\log\frac{1}{\varepsilon}\right)$$

*(see Sec. E.2 of the Appendix) while the local gradient complexity of* `AL2SGD+` *for $\rho = \frac{1}{m}$ and $p = \frac{\lambda}{\lambda+\tilde{L}}$ is*

$$\mathcal{O}\left(\left(m + \sqrt{\frac{m(\tilde{L}+\lambda)}{\mu}}\right)\log\frac{1}{\varepsilon}\right).$$

The communication complexity of `AL2SGD+` is optimal regardless of the relative comparison of $\tilde{L}, \lambda$, which is an improvement over the previous methods. Furthermore, `AL2SGD+` with a slightly different parameters choice enjoys the local gradient complexity which is optimal once $\lambda = \mathcal{O}(\tilde{L})$.

## 5   Experiments

We present empirical evidence to support the theoretical claims of this work. Due to space limitations, we present a fraction of the experiments here only. The remaining plots, as well as the details on the experimental setup, can be found in Section B of the Appendix.

In the first experiment, we study the most practical/ realistic scenario with where the local objective is of a finite-sum structure, while the local oracle provides us with gradients of the summands. In this work, we developed two algorithms capable of dealing with the summand oracle efficiently: `IAPGD+Katyusha` and `AL2SGD+`. We compare both methods against the baseline `L2SGD+` from [19].

The results are presented in Figure 1. In terms of the number of communication rounds, both `AL2SGD+` and `IAPGD+Katyusha` are significantly superior to the `L2SGD+`, as theory predicts. The situation is, however, very different when looking at the local computation. While `AL2SGD+` performs clearly the best, `IAPGD+Katyusha` falls behind `L2SGD`. We presume this happened due to the large constant and $\log$ factors in the local complexity of `IAPGD+Katyusha`.

In the second experiments, we compare two variants of `APGD` presented in Section 4.1: `APGD1` (Algorithm 2) and `APGD2` (Algorithm 3). We consider several synthetic instances of (2) where we vary $\lambda$ and keep remaining parameters (i.e., $L, \mu$) fixed. Our theory predicts that while the rate of `APGD2` should not be influenced by varying $\lambda$, the rate of `APGD1` should grow as $\mathcal{O}(\sqrt{\lambda})$. Similarly, `APGD1` should be favourable if $\lambda \leq L = 1$, while `APGD2` should be the algorithm of choice for $\lambda > L = 1$. As expected, Figure 2 confirms both claims.

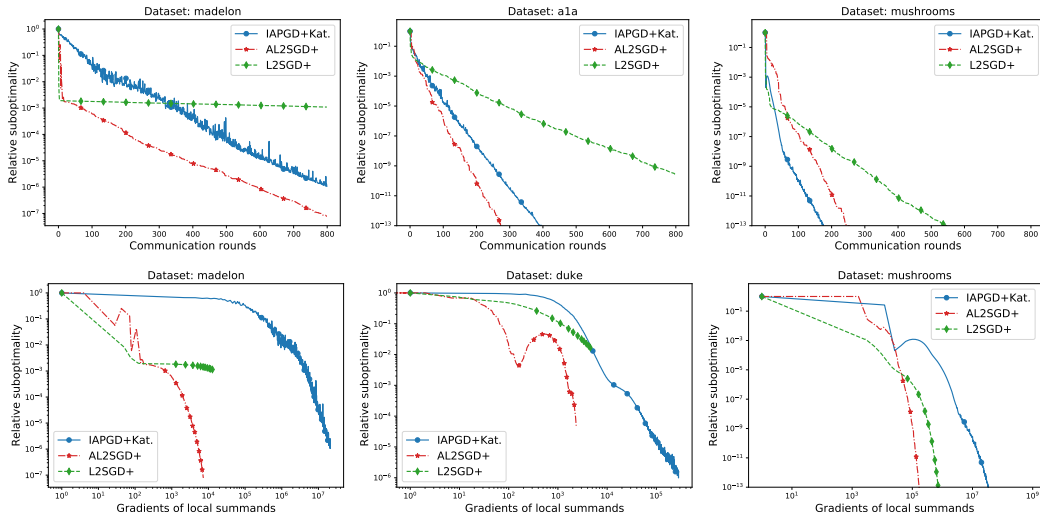

Figure 1: Comparison of `IAPGD+Katyusha`, `AL2SGD+` and `L2SDG+` on logistic regression with `LIBSVM` datasets [7]. Each client owns a random, mutually disjoint subset of the full dataset. First row: communication complexity, second row: local computation complexity for the same experiment.

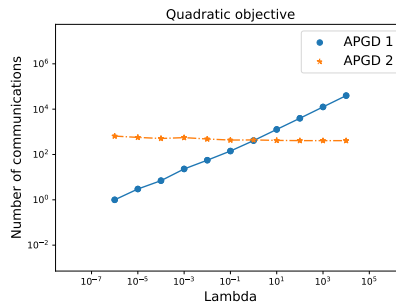

Figure 2: Effect of the parameter $\lambda$ on the communication complexity of `APGD1` and `APGD2`. For each value of $\lambda$ the y-axis indicates the number of communication required to get $10^4$-times closer to the optimum compared to the starting point. Quadratic objective with $n = 50$, $d = 50$.

## Broader Impact

The paper presents lower and upper complexity bounds for the personalized FL formulation (2). While the topic this paper studies—personalized FL—already has a significant societal impact since FL solutions have been and are being deployed in practice, our results are of a theoretical nature. Consequently, the broader impact discussion for this work specifically is not applicable.

## Funding Disclosure

All authors were supported by the KAUST Baseline Research Scheme, and are thankful to support offered by the KAUST Visual Computing Center.

## Footnotes

[1] Up to the stepsize and random number of the local gradient steps.

[2] Surprisingly enough, the non-local algorithms outperform their local counterparts when applied to solve the classical FL formulation (1) with heterogeneous data.

[3] We say that function $h: \mathbb{R}^d \to \mathbb{R}$ is $L$-smooth if for each $z, z' \in \mathbb{R}^d$ we have $h(z) \leq h(z') + \langle \nabla h(z), z' - z \rangle + \frac{L}{2}\|z - z'\|^2$. Similarly, a function $h: \mathbb{R}^d \to \mathbb{R}$ is $\mu$-strongly convex, if for each $z, z' \in \mathbb{R}^d$ it holds $h(z) \geq h(z') + \langle \nabla h(z), z' - z \rangle + \frac{\mu}{2}\|z - z'\|^2$.

[4] Local proximal oracle reveals $\{\text{prox}_{\beta f}(x), \nabla f(x)\}$ for any $x \in \mathbb{R}^{nd}, \beta > 0$. Local gradient oracle reveals $\{\nabla f(x)\}$ for any $x \in \mathbb{R}^{nd}$.

[5]Such an approach was already considered in [28, 40] for the standard FL formulation (1).

[6]See also [52] for a similar lower bound in a slightly different setup.

[7]We suspect that assuming perfect symmetry across nodes is not necessary and can be omitted using more complex arguments. In fact, we believe that allowing for a varying scale of the local problem across the workers so that the condition number remains constant, we can adapt the approach from [21] to obtain the desired local summand gradient complexity without assuming symmetry.

[8]Iterative process (6) is in fact a special case of algorithms proposed in [50]. See Remark 4.1 for details.

[9]The work [50] considers the distributed multi-task learning objective that is more general than (2).

[10]Essentially any accelerated variance reduced algorithm can be used instead of `Katyusha`, for example `ASDCA` [46], `APCG` [29], `Point-SAGA` [9], `MiG` [56], `SAGA-SSNM` [55] and others.

[11]Consequently, we have $\tilde{L} \geq L \geq \frac{\tilde{L}}{m}$.

[12] `L-Katyusha` [42] is a variant of `Katyusha` with a random inner loop length.

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
