[Supplementary Material]

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

[13]Logistic regression loss for on the $j$-th data point $a_j \in \mathbb{R}^d$ is defined as $\phi_j(x) = \log\left(1 + \exp\left(b_j a_j^\top x\right)\right) + \frac{\lambda}{2} \|x\|_2^2$, where $b_j \in \{-1, 1\}$ is the corresponding label.

[14]Relative suboptimality means that for iterates $\left\{x^k\right\}_{k=1}^K$ we plot $\left\{\frac{f(x^k) - f(x^\star)}{f(x^0) - f(x^\star)}\right\}_{k=1}^K$.

[15] See a MatLab symbolic verification at file `eigenvalues.m`.

[16] See a MatLab symbolic verification at file `eigenvaleus.m`.

[17]See Algorithm 1 for the exact meaning.

[18]Inequality $(*)$ holds since for any $0 \leq a < 1$ we have $\frac{-1}{\log(1-a)} \leq \frac{1}{a}$, while $(**)$ holds since $\log\left(\frac{1}{1-\sqrt{\frac{\mu}{\lambda}}}\right) \leq 2\sqrt{\frac{\mu}{\lambda}}$ thanks to $\lambda \geq 2\mu$.

[19]Inequality $(***)$ holds since $\log\left(\frac{1}{1-\sqrt{\frac{\mu}{\lambda}}}\right) \leq 2\sqrt{\frac{\mu}{\lambda}}$ thanks to $\lambda \geq 2\mu$.

[20] Similarly, we could have applied different accelerated variance reduced method with importance sampling such as another version of L-Katyusha [42], for example.

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

# A   Table of frequently used notation

To enhance the reader's convenience when navigating, we here reiterate our notation:

Table 3: Summary of frequently used notation.

| General | | |
|---|---|---|
| $F : \mathbb{R}^{nd} \to \mathbb{R}$ | Global objective | (2) |
| $f_i : \mathbb{R}^n \to \mathbb{R}$ | Local loss on $i$-th node | (2) |
| $x_i \in \mathbb{R}^d$ | Local model on $i$-th node | (2) |
| $x \in \mathbb{R}^{nd}$ | Concatenation of local models $x = [x_1, x_2, \ldots, x_n]$ | (2) |
| $f : \mathbb{R}^{nd} \to \mathbb{R}$ | Average loss over nodes $f(x) \coloneqq 1/n \sum_{i=1}^n f_i(x_i)$ | (2) |
| $\psi : \mathbb{R}^{nd} \to \mathbb{R}$ | Dissimilarity penalty $\psi(x) \coloneqq \frac{1}{2n} \sum_{i=1}^n \|x_i - \bar{x}\|^2$ | (2) |
| $\lambda \geq 0$ | Weight of dissimilarity penalty | (2) |
| $\mathrm{Loc}(x_i, i)$ | local oracle: { proximal, gradient, summand gradient} | Sec. 3 |
| $\mu \geq 0$ | Strong convexity constant of each $f_i$ ($\tilde{f}_{i,j}$) | |
| $L \geq 0$ | Smoothness constant of each $f_i$ | |
| prox | Proximal operator | (4) |
| $m \geq 1$ | Number of local summands of $i$-th local loss $f_i = 1/m \sum_{j=1}^m \tilde{f}_{i,j}$ | Sec. 4.2 |
| $\tilde{f}_{i,j} : \mathbb{R}^n \to \mathbb{R}$ | $j$-th summand of $i$-th local loss, $1 \leq j \leq m$ | Sec. 4.2 |
| $\tilde{L} \geq 0$ | Smoothness constant of each $\tilde{f}_{i,j}$ | Sec. 4.2 |
| $\varepsilon \geq 0$ | Precision | |
| $x^0 \in \mathbb{R}^{nd}$ | Algorithm initialization | |
| $x^\star \in \mathbb{R}^{nd}$ | Optimal solution of (2), $x^\star = [x_1^\star, x_2^\star, \ldots, x_n^\star]$ | |
| $F^\star \in \mathbb{R}$ | Function value at minimum, $F^\star = F(x^\star)$ | |
| Algorithms | | |
| APGD1 | Accelerated Proximal Gradient Descent (Algorithm 2) | Sec. 4.1 |
| APGD2 | Accelerated Proximal Gradient Descent (Algorithm 3) | Sec. 4.1 |
| IAPGD | Inexact Accelerated Proximal Gradient Descent (Algorithm 1) | Sec. 4.2 |
| IAPGD + AGD | IAPGD with AGD as a local sobsolver | Sec. 4.2 |
| IAPGD + Katyusha | IAPGD with Katyusha as a local subsolver | Sec. 4.2 |
| AL2SGD+ | Accelerated Loopless Local Gradient Descent (Algorithm 4) | Sec. 4.3 |
| $p, \rho$ | Probabilities; parameters of AL2SGD+ | Sec. C |

| Dataset | $n$ | $m$ | $d$ | $\lambda$ | $p = \rho$ |
|---|---|---|---|---|---|
| a1a | 5 | 321 | 119 | 0.003 | 0.003 |
| duke | 11 | 4 | 7129 | 0.333 | 0.250 |
| mushrooms | 12 | 677 | 112 | 0.001 | 0.001 |
| madelon | 200 | 10 | 500 | 0.111 | 0.100 |
| phishing | 335 | 33 | 68 | 0.031 | 0.030 |

Table 4: Number of workers and local functions on workers for different datasets for Figures 3 and 4.

## B   Extra experiments and details on the experimental setup

In this section, we provide additional experiments comparing introduced algorithms on logistic regression with LIBSVM data.[13] The local objectives are constructed by evenly dividing to the workers. We vary the parameters $m, n$ among the datasets as specified in Table 4.

We consider two types of assignment of data to the clients: *homogeneous* assignment, where local data are assigned uniformly at random and *heterogeneous* assignment, where we first sort the dataset according to labels, and then assign it to the clients in the given order. The heterogeneous assignment is supposed to better simulate the real-world scenarios. Next, we normalize the data $a_1, a_2, \ldots$, so that $\tilde{f}_{i,j}$ is 1-smooth and set $\mu = 10^{-4}$.

For each dataset we select rather small value of $\lambda$, specifically $\lambda = \frac{1}{m}$. Lastly, for L2SGD+ and AL2SGD+, we choose $p = \rho = 1/m$, which is in the given setup optimal up to a constant factor in terms of the communication. We run the algorithms for $10^3$ communication rounds and track relative suboptimality[14] after each aggregation. Similarly to Figure 1, we plot relative suboptimality agains the number of communication rounds and local gradients computed.

The remaining parameters are selected according to theory for each algorithm with one exception: For IAPGD+Katyusha we run Katyusha as a local subsolver at the iteration $k$ for

$$\sqrt{\frac{m(L + \lambda)}{\mu + \lambda}} + \sqrt{\frac{m\mu(L + \lambda)}{\lambda(\mu + \lambda)}} k$$

iterations (slightly smaller than what our theory suggests).

**Extra experiments**

In the first experiment from this section, we extend the comparison of the stochastic algorithms from Figure 1 to additional datasets. Figure 3 shows a similar behaviour comparing to Figure 1, demonstrating the robustness of our results.

In the second experiment, we investigate the heterogeneous split of the data among the clients mentioned earlier. Figure 4 shows the result. We can see that the data heterogeneity does not influence the convergence significantly and we observe a similar behaviour compared to the homogenous case.

Figure 3: Same as Figure 1, but a different datasets.

Figure 4: Same experiment as Figure 1, but a heterogeneous data split.

# C   Missing parts for Section 4

In this section, we state the algorithms that were mentioned in the main paper: `APGD1` as Algorithm 2, `APGD2` as Algorithm 3 and `AL2SGD+` as Algorithm 4. Next, we state the convergence rates of `APGD1`, `APGD2` as Proposition C.1 and Proposition C.2 respectively. Lastly, we justify (6) via Lemma C.3.

**Proposition C.1** *[5] Let $\{x^k\}_{k=0}^\infty$ be a sequence of iterates generated by Algorithm 2. Then, we have for all $k \geq 0$:*

$$F(x^k) - F^\star \leq \left(1 - \sqrt{\frac{\mu}{\lambda + \mu}}\right)^k \left(F(x^0) - F^\star + \frac{\mu}{2n}\|x^0 - x^\star\|^2\right).$$

---

**Algorithm 2** `APGD1`

---

**Require:** Starting point $y^0 = x^0 \in \mathbb{R}^{nd}$
    **for** $k = 0, 1, 2, \dots$ **do**
        Central server computes the average $\bar{y}^k = \frac{1}{n}\sum_{i=1}^n y_i^k$
        For all clients $i = 1, \dots, n$:
            Solve the regularized local problem $x_i^{k+1} = \operatorname{argmin}_{z \in \mathbb{R}^d} f_i(z) + \frac{\lambda}{2}\|z - \bar{y}^k\|^2$
            Take the momentum step $y_i^{k+1} = x_i^{k+1} + \frac{\sqrt{\lambda} - \sqrt{\mu}}{\sqrt{\lambda} + \sqrt{\mu}}(x_i^{k+1} - x_i^k)$
    **end for**

---

**Proposition C.2** *[5] Let $\{x^k\}_{k=0}^\infty$ be a sequence of iterates generated by Algorithm 3. Then, we have for all $k \geq 0$:*

$$F(x^k) - F^\star \leq \left(1 - \sqrt{\frac{\mu}{L + \mu}}\right)^k \left(F(x^0) - F^\star + \frac{\mu}{2n}\|x^0 - x^\star\|^2\right).$$

---

**Algorithm 3** `APGD2`

---

**Require:** Starting point $y^0 = x^0 \in \mathbb{R}^{nd}$
    **for** $k = 0, 1, 2, \dots$ **do**
        For all clients $i = 1, \dots, n$:
            Take a local gradient step $\tilde{y}_i^k = y_i^k - \frac{1}{L}\nabla f_i(y_i^k)$
        Central server computes the average $\bar{y}^k = \frac{1}{n}\sum_{i=1}^n \tilde{y}_i^k$
        For all clients $i = 1, \dots, n$:
            Take a prox step w.r.t $\lambda\psi$: $x_i^{k+1} = \frac{L\tilde{y}_i^k + \lambda\bar{y}^k}{L + \lambda}$
            Take the momentum step $y_i^{k+1} = x_i^{k+1} + \frac{\sqrt{\frac{L}{\mu}} - 1}{\sqrt{\frac{L}{\mu}} + 1}(x_i^{k+1} - x_i^k)$
    **end for**

---

**Lemma C.3** *Let*

$$x^{k+1} = \operatorname{prox}_{\frac{1}{L_h}\phi}\left(x^k - \frac{1}{L_h}\nabla h(x^k)\right), \tag{8}$$

*for $h(x) := \lambda\psi(x) + \frac{\mu}{2n}\|x\|^2$ and $\phi(x) := f(x) - \frac{\mu}{2n}\|x\|^2$. Then, we have*

$$x_i^{k+1} = \operatorname{prox}_{\frac{1}{\lambda}f_i}(\bar{x}^k).$$

*Further, the iteration complexity of the above process is $\mathcal{O}\left(\frac{\lambda}{\mu}\log\frac{1}{\varepsilon}\right)$.*

**Proof:**

Since function $\psi$ is $\frac{1}{n}$-smooth and $(\nabla\psi(x))_i = \frac{1}{n}(x_i - \bar{x})$ [19], we have $L_h = \frac{\lambda+\mu}{n}, (\nabla h(x))_i = \frac{\lambda}{n}(x_i - \bar{x}) + \frac{\mu}{n}x_i$ and thus

$$x_i^{k+1} = \operatorname*{argmin}_{z\in\mathbb{R}^d} \frac{1}{n}f_i(z) - \frac{\mu}{2n}\|z\|^2 + \frac{\lambda+\mu}{2n}\left\|z - \left(x_i^k - \frac{n}{\lambda+\mu}\left(\frac{\lambda}{n}(x_i^k - \bar{x}^k) + \frac{\mu}{n}x_i^k\right)\right)\right\|^2$$

$$= \operatorname*{argmin}_{z\in\mathbb{R}^d} f_i(z) - \frac{\mu}{2}\|z\|^2 + \frac{\lambda+\mu}{2}\left\|z - \frac{\lambda}{\lambda+\mu}\bar{x}^k\right\|^2$$

$$= \operatorname*{argmin}_{z\in\mathbb{R}^d} f_i(z) + \frac{\lambda}{2}\left\|z - \bar{x}^k\right\|^2 = \operatorname{prox}_{\frac{1}{\lambda}f_i}(\bar{x}^k).$$

Let us now discuss the convergence rate. Given that function $h$ is $\mu_h$-strongly convex, iteration complexity of (4) to reach $\varepsilon$-suboptimality is $\mathcal{O}\left(\frac{L_h}{\mu_h}\log\frac{1}{\varepsilon}\right)$. Since $L_h = \frac{\lambda+\mu}{n}$ (note that $\psi$ is $\frac{1}{n}$ smooth [19]) and $\mu_h = \frac{\mu}{n}$, the iteration complexity of the process (6) becomes $\mathcal{O}\left(\frac{\lambda}{\mu}\log\frac{1}{\varepsilon}\right)$, as desired.

---

**Algorithm 4** AL2SGD+

---

**Require:** $0 < \theta_1, \theta_2 < 1, \eta, \beta, \gamma > 0, \rho, p \in (0,1), y^0 = z^0 = x^0 = w^0 \in \mathbb{R}^{nd}$
  **for** $k = 0, 1, 2, \dots$ **do**
    For all clients $i = 1, \dots, n$:
    $x_i^k = \theta_1 z_i^k + \theta_2 w_i^k + (1 - \theta_1 - \theta_2)y_i^k$
    $\xi = 1$ with probability $p$ and $0$ with probability $1 - p$
    **if** $\xi = 0$ **then**
      For all clients $i = 1, \dots, n$:
      $g_i^k = \frac{1}{n(1-p)}\left(\nabla\tilde{f}_{i,j}(x_i^k) - \nabla\tilde{f}_{i,j}(w_i^k)\right) + \frac{1}{n}\nabla f_i(w_i^k) + \frac{\lambda}{n}(w_i^k - \bar{w}^k)$
      $y_i^{k+1} = x_i^k - \eta g_i^k$
    **else**
      Central server computes the average $\bar{x}^k = \frac{1}{n}\sum_{i=1}^n x_i^k$ and sends it back to the clients
      For all clients $i = 1, \dots, n$:
      $g_i^k = \frac{\lambda}{np}(x_i^k - \bar{x}^k) - \frac{(p^{-1}-1)\lambda}{n}(w_i^k - \bar{w}^k) + \frac{1}{n}\nabla f_i(w_i^k)$
      Set $y_i^{k+1} = x_i^k - \eta g_i^k$
    **end if**
    For all clients $i = 1, \dots, n$:
    $z_i^{k+1} = \beta z_i^k + (1-\beta)x_i^k + \frac{\gamma}{\eta}(y_i^{k+1} - x_i^k)$
    $\xi' = 1$ with probability $\rho$ and $0$ with probability $1 - \rho$
    **if** $\xi' = 0$ **then**
      For all clients $i = 1, \dots, n$:
      $w_i^{k+1} = w_i^k$
    **else**
      For all clients $i = 1, \dots, n$:
      $w_i^{k+1} = y_i^{k+1}$
      Evaluate and store $\nabla f_i(w_i^{k+1})$
      Central server computes the average $\bar{w}^{k+1} = \frac{1}{n}\sum_{i=1}^n w_i^{k+1}$ and sends it back to the clients
    **end if**
  **end for**

---

# D  Proof of Theorem 3.1

In this section, we provide the proof of the Theorem 3.1. In order to do so, we construct a set of function $f_1, f_2, \ldots, f_n$ such that for any algorithm satisfying Assumption 3.1 and the number of the iterations $k$, one must have $\|x^k - x^\star\|^2 \geq \frac{1}{2} \left(1 - 10 \max\left\{\sqrt{\frac{\mu}{\lambda}}, \sqrt{\frac{\mu}{L-\mu}}\right\}\right)^{C(k)+1} \|x^0 - x^\star\|^2$.

Without loss of generality, we consider $x^0 = 0 \in \mathbb{R}^{dn}$. The rationale behind our proof goes as follows: we show that the $nd$–dimensional vector $x^k$ has "a lot of" zero elements while $x^\star$ does not, and hence we might lower bound $\left\|x^k - x^\star\right\|^2$ by $\sum_{j:(x^k)_j=0} (x^\star)_j^2$, which will be large enough. As the main idea of the proof is given, let us introduce our construction.

Let $d = 2T$ for some large $T$ and define the local objectives as follows for even $n$

$$f_1(y) = f_2(y) = \cdots = f_{n/2}(y) \quad := \quad \frac{\mu}{2}\|y\|^2 + ay_1 + \frac{\lambda}{2}c\left(\sum_{i=1}^{T-1}(y_{2i} - y_{2i+1})^2\right) + \frac{\lambda b}{2}y_{2T}^2$$

$$f_{n/2+1}(y) = f_{n/2+2}(y) = \cdots = f_n(y) \quad := \quad \frac{\mu}{2}\|y\|^2 + \frac{\lambda}{2}c\left(\sum_{i=0}^{T-1}(y_{2i+1} - y_{2i+2})^2\right)$$

and as

$$f_1(y) = f_2(y) = \cdots = f_M(y) \quad := \quad \frac{M+1}{M}\frac{\mu}{2}\|y\|^2 + ay_1 + \frac{\lambda}{2}\frac{M+1}{M}c\left(\sum_{i=1}^{T-1}(y_{2i} - y_{2i+1})^2\right) + \frac{\lambda b}{2}y_{2T}^2$$

$$f_{M+1}(y) = f_{M+2}(y) = \cdots = f_n(y) \quad := \quad \frac{\mu}{2}\|y\|^2 + \frac{\lambda}{2}c\left(\sum_{i=0}^{T-1}(y_{2i+1} - y_{2i+2})^2\right)$$

for $n = 2M + 1, M \geq 1$. Note that the smoothness of the objective is now effectively controlled by parameter $c$.

With such definition of functions $f_i(x_i)$, our objective is quadratic and can be written as

$$\frac{n}{\lambda}F(x) = \frac{1}{2}x^\top \mathbf{M}x + \frac{a}{\lambda}x_1, \tag{9}$$

where $\mathbf{M}$ is matrix dependent on parity of $n$. For even $n$, we have

$$\mathbf{M} := \left(\mathbf{I} - \frac{1}{n}ee^\top\right) \otimes \mathbf{I} + \frac{\mu}{\lambda}\mathbf{I} + \begin{pmatrix} \mathbf{M}_1 & 0 \\ 0 & \mathbf{M}_2 \end{pmatrix}, \text{ where}$$

$$\mathbf{M}_1 := \mathbf{I} \otimes \begin{pmatrix} 0 & 0 & 0 & \cdots & 0 \\ 0 & \begin{pmatrix} c & -c \\ -c & c \end{pmatrix} & 0 & & \vdots \\ 0 & 0 & \begin{pmatrix} c & -c \\ -c & c \end{pmatrix} & \ddots & \vdots \\ \vdots & \ddots & & \ddots & \vdots \\ 0 & \cdots & & \cdots & b \end{pmatrix} \text{ and}$$

$$\mathbf{M}_2 := \mathbf{I} \otimes \begin{pmatrix} \begin{pmatrix} c & -c \\ -c & c \end{pmatrix} & 0 & \cdots \\ 0 & \begin{pmatrix} c & -c \\ -c & c \end{pmatrix} & \cdots \\ \vdots & \vdots & \ddots \end{pmatrix}.$$

When $n$ is odd, we have

$$\mathbf{M} := \left(\mathbf{I} - \frac{1}{n}ee^\top\right) \otimes \mathbf{I} + \frac{\mu}{\lambda}\mathbf{I} + \begin{pmatrix} \mathbf{M}_1 + \frac{\mu}{M\lambda}\mathbf{I} & 0 \\ 0 & \mathbf{M}_2 \end{pmatrix}, \text{ where}$$

$$\mathbf{M}_1 := \mathbf{I} \otimes \begin{pmatrix} 0 & 0 & 0 & \cdots & 0 \\ 0 & \begin{pmatrix} \frac{(M+1)c}{M} & -\frac{(M+1)c}{M} \\ -\frac{(M+1)c}{M} & \frac{(M+1)c}{M} \end{pmatrix} & 0 & & \ddots & \vdots \\ 0 & 0 & \begin{pmatrix} \frac{(M+1)c}{M} & -\frac{(M+1)c}{M} \\ -\frac{(M+1)c}{M} & \frac{(M+1)c}{M} \end{pmatrix} & \ddots & \vdots \\ \vdots & \ddots & & \ddots & \vdots \\ 0 & \cdots & & \cdots & \cdots & b \end{pmatrix} \text{ and}$$

$$\mathbf{M}_2 := \mathbf{I} \otimes \begin{pmatrix} \begin{pmatrix} c & -c \\ -c & c \end{pmatrix} & 0 & \cdots \\ 0 & \begin{pmatrix} c & -c \\ -c & c \end{pmatrix} & \cdots \\ \vdots & \vdots & \ddots \end{pmatrix}.$$

Note that our functions $f_k$ depends on parameters $a \in \mathbb{R}, b, c \in \mathbb{R}^+$. We will choose these parameters later in the way that the optimal solution can be obtained easily.

Now let's discuss optimal model for the objective. Since the the objective is strongly convex, the optimum $x^\star$ is unique. Let us find what it is. For the sake of simplicity, denote $y^\star := x_1^\star, z^\star = x_n^\star$. Due to the symmetry, we must have

$$y^\star = x_2^\star = \ldots x_{n/2}^\star, \quad z^\star = x_{n/2+1} = x_{n/2+2} = \ldots x_{n-1}^\star \qquad \text{for even } n$$

and

$$y^\star = x_2^\star = \ldots x_M^\star, \quad z^\star = x_{M+1} = \cdots = x_{n-1}^\star \qquad \text{for odd } n.$$

Now we use the following lemma to express elements of $y^\star, z^\star$ recursively.

**Lemma D.1** *Let*

$$w_i := \begin{cases} \begin{pmatrix} z_i^\star \\ y_i^\star \end{pmatrix} & \text{if } i \text{ is even} \\ \begin{pmatrix} y_i^\star \\ z_i^\star \end{pmatrix} & \text{if } i \text{ is odd} \end{cases}.$$

*Then, we have*

$$w_{i+1} = \mathbf{Q}_r w_i$$

*where*

$$\mathbf{Q}_r := \begin{pmatrix} -\frac{r}{c} & \frac{c + \frac{\mu}{\lambda} + r}{c} \\ -\frac{c + \frac{\mu}{\lambda} + r}{c} & \frac{(c + \frac{\mu}{\lambda} + r)^2}{cr} - \frac{c}{r} \end{pmatrix}$$

*and*

$$r = \begin{cases} \frac{1}{2} & \text{if } n \text{ is even} \\ \frac{M}{n} & \text{if } n \text{ is odd} \end{cases}.$$

To prove the lemma, we shall manipulate the first-order optimality conditions of (9).

**Proof:** For even $n$, the first-order optimality conditions yield

$$\left(c + \frac{1}{2} + \frac{\mu}{\lambda}\right) z_{2i-1}^\star - cz_{2i}^\star - \frac{1}{2}y_{2i-1}^\star = 0 \qquad \text{for } 1 \le i \le T \tag{10}$$

$$\left(c + \frac{1}{2} + \frac{\mu}{\lambda}\right) z_{2i}^\star - cz_{2i-1}^\star - \frac{1}{2}y_{2i}^\star = 0 \qquad \text{for } 1 \le i \le T \tag{11}$$

$$\left(c + \frac{1}{2} + \frac{\mu}{\lambda}\right) y_{2i}^\star - cy_{2i+1}^\star - \frac{1}{2}z_{2i}^\star = 0 \qquad \text{for } 1 \le i \le T - 1 \tag{12}$$

$$\left(c + \frac{1}{2} + \frac{\mu}{\lambda}\right) y_{2i+1}^\star - cy_{2i}^\star - \frac{1}{2}z_{2i+1}^\star = 0 \qquad \text{for } 1 \le i \le T - 1 \tag{13}$$

Equalities (10) and (11) can be equivalently written as

$$\begin{pmatrix} c & 0 \\ -c - r - \frac{\mu}{\lambda} & r \end{pmatrix} \begin{pmatrix} z_{2i}^\star \\ y_{2i}^\star \end{pmatrix} = \begin{pmatrix} c + r + \frac{\mu}{\lambda} & -r \\ -c & 0 \end{pmatrix} \begin{pmatrix} z_{2i-1}^\star \\ y_{2i-1}^\star \end{pmatrix} \qquad \text{for } 1 \le i \le T \tag{14}$$

and consequently we must have for all $1 \le i \le T$

$$\begin{aligned}
\begin{pmatrix} z_{2i}^\star \\ y_{2i}^\star \end{pmatrix} &= \begin{pmatrix} c & 0 \\ -c - r - \frac{\mu}{\lambda} & r \end{pmatrix}^{-1} \begin{pmatrix} c + r + \frac{\mu}{\lambda} & -r \\ -c & 0 \end{pmatrix} \begin{pmatrix} z_{2i-1}^\star \\ y_{2i-1}^\star \end{pmatrix} \\
&= \begin{pmatrix} \frac{c + \frac{\mu}{\lambda} + r}{c} & -\frac{r}{c} \\ \frac{\left(c + \frac{\mu}{\lambda} + r\right)^2}{rc} - \frac{c}{r} & -\frac{c + \frac{\mu}{\lambda} + r}{c} \end{pmatrix} \begin{pmatrix} z_{2i-1}^\star \\ y_{2i-1}^\star \end{pmatrix} \\
&= \mathbf{Q}_r \begin{pmatrix} y_{2i-1}^\star \\ z_{2i-1}^\star \end{pmatrix}.
\end{aligned}$$

Analogously, from (12) and (13) we deduce that for all $1 \le i \le T - 1$

$$\begin{pmatrix} y_{2i+1}^\star \\ z_{2i+1}^\star \end{pmatrix} = \mathbf{Q}_r \begin{pmatrix} z_{2i}^\star \\ y_{2i}^\star \end{pmatrix}.$$

**For odd** $n$, the first-order optimality conditions yield

$$\left(c + \frac{M}{n} + \frac{\mu}{\lambda}\right) z_{2i-1}^\star - cz_{2i}^\star - \frac{M}{n}y_{2i-1}^\star = 0 \qquad \text{for } 1 \le i \le T \tag{15}$$

$$\left(c + \frac{M}{n} + \frac{\mu}{\lambda}\right) z_{2i}^\star - cz_{2i-1}^\star - \frac{M}{n}y_{2i}^\star = 0 \qquad \text{for } 1 \le i \le T \tag{16}$$

$$\left(\frac{M+1}{M}c + \frac{M+1}{n} + \frac{M+1}{M}\frac{\mu}{\lambda}\right) y_{2i}^\star - \frac{M+1}{M}cy_{2i+1}^\star - \frac{M+1}{n}z_{2i}^\star = 0 \qquad \text{for } 1 \le i \le T - 1 \tag{17}$$

$$\left(\frac{M+1}{M}c + \frac{M+1}{n} + \frac{M+1}{M}\frac{\mu}{\lambda}\right) y_{2i+1}^\star - \frac{M+1}{M}cy_{2i}^\star - \frac{M+1}{n}z_{2i+1}^\star = 0 \qquad \text{for } 1 \le i \le T - 1 \tag{18}$$

Equalities (15) and (16) can be equivalently written as

$$\begin{pmatrix} c & 0 \\ -c - r - \frac{\mu}{\lambda} & r \end{pmatrix} \begin{pmatrix} z_{2i}^\star \\ y_{2i}^\star \end{pmatrix} = \begin{pmatrix} c + r + \frac{\mu}{\lambda} & -r \\ -c & 0 \end{pmatrix} \begin{pmatrix} z_{2i-1}^\star \\ y_{2i-1}^\star \end{pmatrix} \qquad \text{for } 1 \le i \le T,$$

which is identical to (14), and thus

$$\begin{pmatrix} z_{2i}^\star \\ y_{2i}^\star \end{pmatrix} = \mathbf{Q}_r \begin{pmatrix} y_{2i-1}^\star \\ z_{2i-1}^\star \end{pmatrix}.$$

Similarly, (17) and (18) imply that for all $1 \le i \le T - 1$

$$\begin{pmatrix} y_{2i+1}^\star \\ z_{2i+1}^\star \end{pmatrix} = \mathbf{Q}_r \begin{pmatrix} z_{2i}^\star \\ y_{2i}^\star \end{pmatrix}.$$

As consequence of Lemma D.1, we have that $w_k = \mathbf{Q}_r^{k-1}w_1$ with $\frac{1}{3} \leq r \leq \frac{1}{2}$. Now we use the flexibility to choose $a \in \mathbb{R}, b \in \mathbb{R}^+$, so that we can find suitable $w_k$ (and thus suitable $x^\star$). Specifically, we aim to choose $a, b$, so that $w_1$ will be the eigenvector of $\mathbf{Q}_r$, corresponding to a suitable eigenvalue $\gamma$ of matrix $\mathbf{Q}_r$. Then $w_k$ could be written as $w_k = \gamma^k w_1$.

**Lemma D.2** *Choose* $c := \begin{cases} 1 & \text{if } L \geq \lambda + \mu \\ \delta\frac{\mu}{\lambda}, \delta \geq 1 & \text{if } L < \lambda + \mu \end{cases}$ *and*

$$b := \begin{cases} \frac{\frac{\mu^2}{\lambda^2}+2\frac{\mu}{\lambda}+2r+2r\frac{\mu}{\lambda}+2r^2+\left(\frac{\mu}{\lambda}(\frac{\mu}{\lambda}+2r)(\frac{\mu}{\lambda}+2)(\frac{\mu}{\lambda}+2r+2)\right)^{\frac{1}{2}}}{2r(1+\frac{\mu}{\lambda}+r)} - 1 - \frac{\mu}{\lambda} & \text{if } L \geq \lambda + \mu \\ \frac{\frac{\mu^2}{\lambda^2}+2r^2+2r\frac{\mu}{\lambda}+2\delta r\frac{\mu}{\lambda}+2\delta\frac{\mu^2}{\lambda^2}+\frac{\mu}{\lambda}\left((2\delta+1)(\frac{\mu}{\lambda}+2r)(\frac{\mu}{\lambda}+2r+2\delta\frac{\mu}{\lambda})\right)^{\frac{1}{2}}}{2r(\frac{\mu}{\lambda}+r+\delta\frac{\mu}{\lambda})} - 1 - \frac{\mu}{\lambda} & \text{if } L < \lambda + \mu \end{cases}. \quad (19)$$

*Then, we have $b \geq 0$ and*

$$w_i = \gamma^{i-1}w_1 \neq \begin{pmatrix} 0 \\ 0 \end{pmatrix} \qquad \text{for } i = 1, 2, \ldots, d$$

*where*

$$\gamma := \begin{cases} \frac{\frac{\mu^2}{\lambda^2}+2\frac{\mu}{\lambda}+2r+2r\frac{\mu}{\lambda}-\left(\frac{\mu}{\lambda}(\frac{\mu}{\lambda}+2r)(\frac{\mu}{\lambda}+2)(\frac{\mu}{\lambda}+2r+2)\right)^{\frac{1}{2}}}{2r} \geq 1 - 10\sqrt{\frac{\mu}{\lambda}} & \text{if } L \geq \lambda + \mu \\ \frac{\frac{\mu}{\lambda}+2r+2\delta r+2\delta\frac{\mu}{\lambda}-\left((2\delta+1)(\frac{\mu}{\lambda}+2r)(\frac{\mu}{\lambda}+2r+2\delta\frac{\mu}{\lambda})\right)^{\frac{1}{2}}}{2\delta r} \geq 1 - 10\sqrt{\frac{1}{\delta}} & \text{if } L < \lambda + \mu \end{cases}. \quad (20)$$

**Proof:** First, note that if $c = 1$, each local objective is $(\mu + \lambda)$-smooth, and thus also $L$-smooth (and therefore the choice of $c$ does not contradict the smoothness). Next, if $L \geq \lambda + \mu$, the vector

$$v := \begin{pmatrix} \frac{\frac{\mu^2}{\lambda^2}+2\frac{\mu}{\lambda}+2r+2r\frac{\mu}{\lambda}+2r^2+\left(\frac{\mu}{\lambda}(\frac{\mu}{\lambda}+2r)(\frac{\mu}{\lambda}+2)(\frac{\mu}{\lambda}+2r+2)\right)^{\frac{1}{2}}}{2r(1+\frac{\mu}{\lambda}+r)} \\ 1 \end{pmatrix}$$

is an unnormalized eigenvector of $\mathbf{Q}_r$ corresponding to eigenvalue $\gamma$.[15] Next, we prove $b \geq 0$ and $\gamma \geq 1 - 10\sqrt{\frac{\mu}{\lambda}}$ using Mathematica, see the file `proof.nb` and the screen shot below.

```
In[1]:= FindInstance[
    (2 * x + 2 * r + 2 * r * x + x ^ 2 - ((x) * (x + 2 * r) * (x + 2) * (x + 2 + 2 * r)) ^ (0.5)) / (2 * r) -
        1 + 10 * x ^ (0.5) < 0 && x > 0 && x ≤ 1 && r ≥ 1 / 3 && r ≤ 1 / 2, {x, r}]

Out[1]= {}

In[2]:= FindInstance[
    (2 * x + 2 * r + 2 * r * x + x ^ 2 + 2 * r ^ 2 + ((x) * (x + 2 * r) * (x + 2) * (x + 2 + 2 * r)) ^ (0.5)) /
        (2 * r * (1 + x + r)) - 1 - x < 0 && x > 0 && x ≤ 1 && r ≥ 1 / 3 && r ≤ 1 / 2, {x, r}]

Out[2]= {}
```

Let us look now at the case where $L \leq \lambda + \mu$. Now, the vector

$$v := \begin{pmatrix} \frac{\frac{\mu^2}{\lambda^2}+2r^2+2r\frac{\mu}{\lambda}+2\delta r\frac{\mu}{\lambda}+2\delta\frac{\mu^2}{\lambda^2}+\frac{\mu}{\lambda}\left((2\delta+1)(\frac{\mu}{\lambda}+2r)(\frac{\mu}{\lambda}+2r+2\delta\frac{\mu}{\lambda})\right)^{\frac{1}{2}}}{2r(\frac{\mu}{\lambda}+r+\delta\frac{\mu}{\lambda})} \\ 1 \end{pmatrix}$$

is an unnormalized eigenvector of $\mathbf{Q}_r$ corresponding to eigenvalue $\gamma$.[16] Next, we prove $b \geq 0$ and $\gamma \geq 1 - 10\sqrt{\frac{1}{\delta}}$ using Mathematica, see the file `proof.nb` and the screen shot below.

```
In[12]:= FindInstance[
         (x + 2 * r + 2 * delta * r + 2 * delta * x - ((2 * delta + 1) * (x + 2 * r) * (x + 2 * r + 2 * delta * x)) ^ (0.5)) /
            (2 * delta * r) - 1 + 10 / (delta ^ (0.5)) < 0 && x > 0 && x ≤ 1 && r ≥ 1/3  && r ≤ 1/2 &&
          delta ≥  1 && delta ≤ 1/x, {x, r, delta}]

Out[12]= {}

In[13]:= FindInstance[
         (x ^ 2 + 2 * r ^ 2 + 2 * r * x +  2 * delta * r * x + 2 * delta * x ^ 2 + x * ((2 * delta + 1) * (x + 2 * r)
                     * (x + 2) * (x + 2 * r + 2 * delta * x)) ^ (0.5)) / (2 * r * (x + r + delta * x)) - 1 - x < 0 &&
          x > 0 && x ≤ 1 && r ≥ 1/3  && r ≤ 1/2 && delta ≥  1 && delta ≤ 1/x, {x, r, delta}]

Out[13]= {}
```

Setting $b$ according to (19) we assure that $w_i$ is a multiple of $v$ and consequently we have

$$w_i = \gamma^{i-1} w_1 \qquad \text{for } i = 1, 2, \dots, d$$

as desired. It remains to mention that $w_i \neq \begin{pmatrix} 0 \\ 0 \end{pmatrix}$ regardless of the choice of $a \neq 0$.

**Proof: Theorem 3.1**

Let $x^0 = 0 \in \mathbb{R}^{nd}$. Note that our oracle allows us at most $K + 1$ nonzero coordinates of $x^K$ after $K$ rounds of communications. Consequently,

$$
\begin{aligned}
\frac{\|x^K - x^\star\|^2}{\|x^0 - x^\star\|^2} &\geq \frac{1}{2} \frac{\sum_{j=K+2}^d \|w_j\|^2}{\sum_{j=1}^d \|w_j\|^2} = \frac{1}{2} \frac{\sum_{j=K+2}^d \gamma^{j-1} \|w_1\|^2}{\sum_{j=1}^d \gamma^{j-1} \|w_1\|^2} = \frac{1}{2} \frac{\gamma^{K+1} \sum_{j=0}^{d-K-2} \gamma^j}{\sum_{j=0}^{d-1} \gamma^j} \\
&= \frac{1}{2} \gamma^{K+1} \frac{1 - \gamma^{d-K-1}}{1 - \gamma^d} \overset{(*)}{\geq} \frac{1}{4} \left( 1 - 10 \max \left\{ \sqrt{\frac{\mu}{\lambda}}, \sqrt{\frac{1}{\delta}} \right\} \right)^{K+1} \\
&= \frac{1}{4} \left( 1 - 10 \max \left\{ \sqrt{\frac{\mu}{\lambda}}, \sqrt{\frac{\mu}{L - \mu}} \right\} \right)^{K+1}
\end{aligned}
$$

where the inequality $(*)$ holds for large enough $T$ (and consequently large enough $d = 2T$). $\qquad \square$

# E  Proofs for Section 4

## E.1  Towards the Proof of Theorems 4.2 and 4.3

**Proposition E.1** *Iterates of Algorithm 1 satisfy*

$$F(x^k) - F^\star$$

$$\leq \left(1 - \sqrt{\frac{\mu}{\lambda}}\right)^k \left(\sqrt{2(F(x^0) - F^\star)} + 2\sqrt{\frac{\lambda}{\mu}} \left(\sum_{i=1}^k \epsilon_i^{\frac{1}{2}} \left(1 - \sqrt{\frac{\mu}{\lambda}}\right)^{-\frac{i}{2}}\right) + \sqrt{\sum_{i=1}^k \epsilon_i \left(1 - \sqrt{\frac{\mu}{\lambda}}\right)^{-i}}\right)^2.$$

(21)

**Proof:** First, notice that the objective is $\frac{\lambda}{n}$ smooth and $\frac{\mu}{n}$-strongly convex. Next, the error in the evaluation of the proximal operator at iteration $k$ can be expressed as

$$\sum_{i=1}^n \frac{1}{n} f_i(x_i^{k+1}) + \frac{\lambda}{2n} \|x_i^{k+1} \bar{y}^k\|^2 \leq \sum_{i=1}^n \frac{1}{n} \epsilon_k = \epsilon_k.$$

It remains to apply Proposition 4 from [45].

### E.1.1  General convergence rate of `IAPGD`

Theorem E.2 shows that the expected number of communications that Algorithm 1 requires to reach $\varepsilon$-approximate solution is $\tilde{\mathcal{O}}\left(\sqrt{\frac{\lambda}{\mu}}\right)$, given that (22) holds.

**Theorem E.2** *Assume that for all $k \geq 0, 1 \leq i \leq n$, the subproblem (7) was solved up to a suboptimality[17] $\epsilon_k$ by a possibly randomized iterative algorithm such that*

$$\mathbb{E}\left[\epsilon_k \mid x^k\right] \leq \left(1 - \sqrt{\frac{\mu}{\lambda}}\right)^{2k} R^2$$

(22)

*for some fixed $R > 0$. Consequently, we have*

$$\mathbb{E}\left[(F(x^k) - F^\star)^{\frac{1}{2}}\right] \leq \left(1 - \sqrt{\frac{\mu}{\lambda}}\right)^{\frac{k}{2}} \left(\sqrt{2(F(x^0) - F^\star)} + 2\left(2\sqrt{\frac{\lambda}{\mu}} + 1\right)\sqrt{\frac{\lambda}{\mu}} R\right).$$

(23)

**Proof:**
Let $\omega := 1 - \sqrt{\frac{\mu}{\lambda}}$. Proposition E.1 gives us

$$\left(F(x^k) - F^\star\right)^{\frac{1}{2}} \overset{(21)}{\leq} \omega^{\frac{k}{2}} \left(\sqrt{2(F(x^0) - F^\star)} + 2\sqrt{\frac{\lambda}{\mu}} \left(\sum_{i=1}^k \epsilon_i^{\frac{1}{2}} \omega^{-\frac{i}{2}}\right) + \sqrt{\sum_{i=1}^k \epsilon_i \omega^{-i}}\right)$$

$$\leq \omega^{\frac{k}{2}} \left(\sqrt{2(F(x^0) - F^\star)} + \left(2\sqrt{\frac{\lambda}{\mu}} + 1\right)\left(\sum_{i=1}^k \epsilon_i^{\frac{1}{2}} \omega^{-\frac{i}{2}}\right)\right).$$

Taking the expectation, we get

$$\mathbb{E}\left[\left(F(x^k) - F^\star\right)^{\frac{1}{2}}\right] \leq \omega^{\frac{k}{2}}\left(\sqrt{2(F(x^0) - F^\star)} + \left(2\sqrt{\frac{\lambda}{\mu}} + 1\right)\left(\sum_{i=1}^{k}\mathbb{E}\left[\epsilon_i^{\frac{1}{2}}\right]\omega^{-\frac{i}{2}}\right)\right)$$

$$\leq \omega^{\frac{k}{2}}\left(\sqrt{2(F(x^0) - F^\star)} + \left(2\sqrt{\frac{\lambda}{\mu}} + 1\right)\left(\sum_{i=1}^{k}\mathbb{E}\left[\epsilon_i\right]^{\frac{1}{2}}\omega^{-\frac{i}{2}}\right)\right)$$

$$\overset{(22)}{\leq} \omega^{\frac{k}{2}}\left(\sqrt{2(F(x^0) - F^\star)} + \left(2\sqrt{\frac{\lambda}{\mu}} + 1\right)R\left(\sum_{i=1}^{k}\omega^{\frac{i}{2}}\right)\right)$$

$$\leq \omega^{\frac{k}{2}}\left(\sqrt{2(F(x^0) - F^\star)} + \left(2\sqrt{\frac{\lambda}{\mu}} + 1\right)R\left(\sum_{i=1}^{\infty}\omega^{\frac{i}{2}}\right)\right)$$

$$= \omega^{\frac{k}{2}}\left(\sqrt{2(F(x^0) - F^\star)} + \left(2\sqrt{\frac{\lambda}{\mu}} + 1\right)R\frac{\omega^{\frac{1}{2}}}{1 - \omega^{\frac{1}{2}}}\right)$$

$$\leq \omega^{\frac{k}{2}}\left(\sqrt{2(F(x^0) - F^\star)} + \left(2\sqrt{\frac{\lambda}{\mu}} + 1\right)2R\frac{1}{1 - \omega}\right)$$

$$= \omega^{\frac{k}{2}}\left(\sqrt{2(F(x^0) - F^\star)} + \left(2\sqrt{\frac{\lambda}{\mu}} + 1\right)2R\sqrt{\frac{\lambda}{\mu}}\right),$$

which is exactly (23).

### E.1.2 Proof of Theorem 4.2

Denote $\mathcal{S}' := \{x; F(x) \leq F^\star + 8(F(x^0) - F^\star)\}$, $\mathcal{S} := \{(2-\alpha)x' - (1-\alpha)x''; x', x'' \in \mathcal{S}', 0 \leq \alpha \leq 1\}$ and $D := \mathrm{Diam}(\mathcal{S}) < \infty$. Consequently,

$$D^2 \leq 36\max_{x \in \mathcal{S}}\|x - x^\star\|^2 \leq \frac{18n}{\mu}\max_{x \in \mathcal{S}}(F(x) - F(x^\star)) \leq \frac{144n}{\mu}(F(x^0) - F^\star) \qquad (24)$$

Let us proceed with induction. Suppose that for all $0 \leq t < k$ we have

$$F(x^i) - F^\star \leq 8\left(1 - \sqrt{\frac{\mu}{\lambda}}\right)^t (F(x^0) - F^\star).$$

Consequently, $x^t \in \mathcal{S}'$ for all $0 \leq t < k$. Thanks to the update rule of sequence $\{y\}_{t=1}^{\infty}$, we must have $y^{k-1} \in \mathcal{S}$. Next, define $\hat{x}_i^k := \mathrm{argmin}_{z \in \mathbb{R}^d} f_i(z) + \frac{\lambda}{2n}\|z - \bar{y}^{k-1}\|^2$. Clearly, $\hat{x}^k \in \mathcal{S}$, and consequently, $\|\hat{x}^k - y^{k-1}\|^2 \leq D^2$.

We will next show that

$$\epsilon_k \leq R^2\omega^{2k}, \qquad (25)$$

where

$$R := \frac{\sqrt{2(F(x^0) - F^\star)}}{2\sqrt{\frac{\lambda}{\mu}}\left(2\sqrt{\frac{\lambda}{\mu}} + 1\right)}, \qquad \omega := 1 - \sqrt{\frac{\mu}{\lambda}}. \qquad (26)$$

Define $h_i^k(z) := f_i(z) + \frac{\lambda}{2n}\|z - \bar{y}^{k-1}\|^2$. Since $h_i^k$ is $\frac{1}{n}(L+\lambda)$ smooth and $\frac{1}{n}(\mu+\lambda)$ strongly convex, running AGD locally for $c_1 + c_2 k$ iterations with[18]

$$c_1 \quad := \quad -\frac{\log \frac{4LD^2}{R^2}}{\log\left(1 - \sqrt{\frac{\mu+\lambda}{L+\lambda}}\right)} \overset{(*)}{\leq} \sqrt{\frac{L+\lambda}{\mu+\lambda}} \log \frac{4LD^2}{R^2} \overset{(24)}{\leq} \sqrt{\frac{L+\lambda}{\mu+\lambda}} \log \frac{1152 L\lambda n \left(2\sqrt{\frac{\lambda}{\mu}}+1\right)^2}{\mu^2},$$

$$c_2 \quad := \quad \frac{2\log\omega}{\log\left(1 - \sqrt{\frac{\mu+\lambda}{L+\lambda}}\right)} \overset{(*)+(**)}{\leq} 4\sqrt{\frac{\mu(L+\lambda)}{\lambda(\mu+\lambda)}}$$

yields

$$\epsilon_k \overset{\text{[45], Prop 4}}{\leq} \left(1 - \sqrt{\frac{\mu+\lambda}{L+\lambda}}\right)^{c_1+c_2 k} 4\left(\sum_{i=i}^{n}\left(h_i^k(y_i^{k-1}) - h_i^k(\hat{x}_i^k)\right)\right)$$

$$\leq \left(1 - \sqrt{\frac{\mu+\lambda}{L+\lambda}}\right)^{c_1+c_2 k} 4LD^2$$

$$= \exp\left(c_2 k \log\left(1 - \sqrt{\frac{\mu+\lambda}{L+\lambda}}\right) + c_1 \log\left(1 - \sqrt{\frac{\mu+\lambda}{L+\lambda}}\right) + \log\left(4LD^2\right)\right)$$

$$= \exp\left(2k\log\omega + \log(R^2)\right)$$

$$= R^2\omega^{2k}$$

as desired.

Next, Theorem E.2 gives us

$$F(x^k) - F^\star \overset{(23)}{\leq} \left(1 - \sqrt{\frac{\mu}{\lambda}}\right)^k \left(\sqrt{2(F(x^0) - F^\star)} + 2\left(2\sqrt{\frac{\lambda}{\mu}}+1\right)\sqrt{\frac{\lambda}{\mu}}R\right)^2$$

$$\overset{(26)}{=} 8\left(1 - \sqrt{\frac{\mu}{\lambda}}\right)^k (F(x^0) - F^\star),$$

as desired.

Consequently, in order to reach $\varepsilon$ suboptimality, we shall set $k = \mathcal{O}\left(\sqrt{\frac{\lambda}{\mu}}\log\frac{1}{\varepsilon}\right)$. The total number of local gradient computation thus is

$$\sum_{i=1}^{k}(c_1 + c_2 i) \quad = \quad kc_1 + c_2\mathcal{O}(k^2)$$

$$= \quad \mathcal{O}\left(\sqrt{\frac{L+\lambda}{\mu+\lambda}}\log\frac{32L\lambda n^2\left(4\sqrt{\frac{\lambda}{\mu}}+1\right)^2}{\mu^2}\sqrt{\frac{\lambda}{\mu}}\log\frac{1}{\varepsilon} + \sqrt{\frac{\mu(L+\lambda)}{\lambda(\mu+\lambda)}}\frac{\lambda}{\mu}\left(\log\frac{1}{\varepsilon}\right)^2\right)$$

$$= \quad \mathcal{O}\left(\sqrt{\frac{L+\lambda}{\mu}}\log\frac{1}{\varepsilon}\left(\log\frac{L\lambda n}{\mu} + \log\frac{1}{\varepsilon}\right)\right).$$

$\square$

### E.1.3 Proof of Theorem 4.3

Next, since the sequence of iterates $\{x^k\}_{k=0}^{\infty}$ is bounded, so is the sequence $\{y^k\}_{k=0}^{\infty}$, and consequently, the initial distance to the optimum is bounded for each local subproblem too. As the local objective is $(\tilde{L} + \lambda)$-smooth and $(\mu + \lambda)$-strongly convex, in order to guarantee (22), `Katyusha` requires

$$
\mathcal{O}\left(\left(m + \sqrt{m\frac{\tilde{L}+\lambda}{\mu+\lambda}}\right)\log\frac{1}{R^2\omega^2 k}\right) = \mathcal{O}\left(\left(m + \sqrt{m\frac{\tilde{L}+\lambda}{\mu+\lambda}}\right)\left(\log\frac{1}{R^2} + 2k\log\frac{1}{\omega}\right)\right)
$$

$$
\overset{(***)}{=} \mathcal{O}\left(\left(m + \sqrt{m\frac{\tilde{L}+\lambda}{\mu+\lambda}}\right)\left(\log\frac{1}{R^2} + k\sqrt{\frac{\mu}{\lambda}}\right)\right)
$$

iterations.[19]

Lastly, since `Katyusha` requires $\mathcal{O}(1)$ local stochastic gradient evaluations on average, the total local gradient complexity becomes

$$
\sum_{t=1}^{\tilde{\mathcal{O}}\left(\sqrt{\frac{\lambda}{\mu}}\right)} \mathcal{O}\left(\left(m + \sqrt{m\frac{\tilde{L}+\lambda}{\mu+\lambda}}\right)\left(\log\frac{1}{R^2} + t\sqrt{\frac{\mu}{\lambda}}\right)\right) = \tilde{\mathcal{O}}\left(\left(m\sqrt{\frac{\lambda}{\mu}} + \sqrt{m\frac{\tilde{L}+\lambda}{\mu}}\right)\right).
$$

$\square$

## E.2 Towards the Proof of Theorem 4.4

**Lemma E.3** *Suppose that $\tilde{f}_{ij}$ is $\tilde{L}$ smooth for all $1 \leq i \leq n, 1 \leq j \leq m$. Let $g^k$ be a variance reduced stochastic gradient estimator from Algorithm 4 and define*

$$
\mathcal{L} := \max\left\{\frac{\tilde{L}}{n(1-p)}, \frac{\lambda}{np}\right\}.
$$

*Then, we have*

$$
\mathbb{E}\left[\left\|g^k - \nabla F(x^k)\right\|^2\right] \leq 2\mathcal{L}D_F(w^k, x^k). \tag{27}
$$

**Proof:**

$$\mathbb{E}\left[\left\|g^k - \nabla F(x^k)\right\|^2\right] \;=\; \frac{1-p}{m}\sum_{j=1}^{m}\sum_{i=1}^{n}\left\|\frac{1}{1-p}\left(\nabla\tilde{f}_{ij}(x^k) - \nabla\tilde{f}_{ij}(w^k)\right) - \left(\nabla F(x^k) - \nabla F(w^k)\right)\right\|^2$$

$$+ p\left\|\frac{\lambda}{p}\left(\nabla\psi(x^k) - \nabla\psi(w^k)\right) - \left(\nabla F(x^k) - \nabla F(w^k)\right)\right\|^2$$

$$\leq\; \frac{1-p}{m}\sum_{j=1}^{m}\sum_{i=1}^{n}\left\|\frac{1}{1-p}\left(\nabla\tilde{f}_{ij}(x^k) - \nabla\tilde{f}_{ij}(w^k)\right)\right\|^2$$

$$+ p\left\|\frac{\lambda}{p}\left(\nabla\psi(x^k) - \nabla\psi(w^k)\right)\right\|^2$$

$$=\; \frac{1}{m(1-p)}\sum_{j=1}^{m}\sum_{i=1}^{n}\left\|\nabla\tilde{f}_{ij}(x^k) - \nabla\tilde{f}_{ij}(w^k)\right\|^2 + \frac{\lambda^2}{p}\left\|\left(\nabla\psi(x^k) - \nabla\psi(w^k)\right)\right\|^2$$

$$\overset{(*)}{\leq}\; \frac{2\tilde{L}}{nm(1-p)}\sum_{j=1}^{m}\sum_{i=1}^{n}D_{\tilde{f}_{ij}}(w^k,x^k) + \frac{2\lambda^2}{np}D_{\psi}(w^k,x^k)$$

$$=\; \frac{2\tilde{L}}{1-p}D_f(w^k,x^k) + \frac{2\lambda^2}{np}D_{\psi}(w^k,x^k)$$

$$\leq\; 2\max\left\{\frac{\tilde{L}}{n(1-p)}, \frac{\lambda}{np}\right\}D_F(w^k,x^k)$$

$$=\; 2\mathcal{L}D_F(w^k,x^k).$$

Above, $(*)$ holds since $\tilde{f}$ is $\frac{L}{n}$ smooth and $\psi$ is $\frac{1}{n}$ smooth [19].

**Proposition E.4** *Let $\tilde{f}_{ij}$ be $L$ smooth and $\mu$ strongly convex for all $1 \leq i \leq n, 1 \leq j \leq m$. Define the following Lyapunov function:*

$$\Psi^k \;:=\; \left\|z^k - x^\star\right\|^2 + \frac{2\gamma\beta}{\theta_1}\left[F(y^k) - F(x^\star)\right] + \frac{(2\theta_2 + \theta_1)\gamma\beta}{\theta_1\rho}\left[F(w^k) - F(x^\star)\right],$$

*and let*

$$L_F \;=\; \frac{1}{n}(\lambda + \tilde{L}),$$

$$\eta \;=\; \frac{1}{4}\max\{L_F, \mathcal{L}\}^{-1},$$

$$\theta_2 \;=\; \frac{\mathcal{L}}{2\max\{L_F, \mathcal{L}\}},$$

$$\gamma \;=\; \frac{1}{\max\{2\mu/n, 4\theta_1/\eta\}},$$

$$\beta \;=\; 1 - \frac{\gamma\mu}{n} \quad \text{and}$$

$$\theta_1 \;=\; \min\left\{\frac{1}{2}, \sqrt{\frac{\eta\mu}{n}\max\left\{\frac{1}{2}, \frac{\theta_2}{\rho}\right\}}\right\}.$$

*Then the following inequality holds:*

$$\mathbb{E}\left[\Psi^{k+1}\right] \leq \left[1 - \frac{1}{4}\min\left\{\rho, \sqrt{\frac{\mu}{2n\max\left\{L_F, \frac{\mathcal{L}}{\rho}\right\}}}\right\}\right]\Psi^0.$$

*As a consequence, iteration complexity of Algorithm 4 is*

$$\mathcal{O}\left(\left(\frac{1}{\rho} + \sqrt{\frac{\max\left\{\frac{\tilde{L}}{1-p}, \frac{\lambda}{p}\right\}}{\rho\mu}}\right)\log\frac{1}{\varepsilon}\right).$$

At the same time, the communication complexity of AL2SGD+ is

$$\mathcal{O}\left((\rho + p(1-p))\left(\frac{1}{\rho} + \sqrt{\frac{\max\left\{\frac{\tilde{L}}{1-p}, \frac{\lambda}{p}\right\}}{\rho\mu}}\right)\log\frac{1}{\varepsilon}\right)$$

and the local stochastic gradient complexity is

$$\mathcal{O}\left((\rho m + (1-\rho))\left(\frac{1}{\rho} + \sqrt{\frac{\max\left\{\frac{\tilde{L}}{1-p}, \frac{\lambda}{p}\right\}}{\rho\mu}}\right)\log\frac{1}{\varepsilon}\right).$$

**Proof:**

Note that AL2SGD+ is a special case of L-Katyusha from [18].[20] In order to apply Theorem 4.1 therein directly, it suffices to notice that function $F$ is $L_F = \frac{1}{n}(\lambda + \tilde{L})$ smooth and $\frac{1}{n}\mu$ strongly convex, and at the same time, thanks to Lemma E.3 we have

$$\mathbb{E}\left[\left\|g^k - \nabla F(x^k)\right\|^2\right] \le 2\mathcal{L}D_F(w^k, x^k).$$

Connsequently, we immediately get the iteration complexity. The local stochastic gradient complexity of a single iteration of AL2SGD+ is 0 if $\xi = 1, \xi' = 0$, 1 if $\xi = 0, \xi' = 0$, $m$ if $\xi = 0, \xi' = 1$ and $m + 1$ if $\xi = 1, \xi' = 1$. Thus, the total expected local stochastic gradient complexity is bounded by

$$\mathcal{O}\left((\rho m + (1-\rho))\left(\frac{1}{\rho} + \sqrt{\frac{\max\left\{\frac{\tilde{L}}{1-p}, \frac{\lambda}{p}\right\}}{\rho\mu}}\right)\log\frac{1}{\varepsilon}\right)$$

as desired. Next, the total communication complexity is bounded by the sum of the communication complexities coming from the full gradient computation (if statement that includes $\xi$) and the rest (if statement that includes $\xi'$). The former requires a communication if $\xi' = 1$, the latter if two consecutive $\xi$-coin flips are different (see [19]), yielding the expected total communication $\mathcal{O}(\rho + p(1-p))$ per iteration.

### E.2.1 Proof of Theorem 4.4

For $\rho = p(1-p)$ and $p = \frac{\lambda}{\lambda + \tilde{L}}$, the total communication complexity of AL2SGD+ becomes

$$\mathcal{O}\left((\rho + p(1-p))\left(\frac{1}{\rho} + \sqrt{\frac{\max\left\{\frac{\tilde{L}}{1-p}, \frac{\lambda}{p}\right\}}{\rho\mu}}\right)\log\frac{1}{\varepsilon}\right) = \mathcal{O}\left(\sqrt{\frac{p\tilde{L} + (1-p)\lambda}{\mu}}\log\frac{1}{\varepsilon}\right)$$

$$= \mathcal{O}\left(\sqrt{\frac{\tilde{L}\lambda}{(\tilde{L} + \lambda)\mu}}\log\frac{1}{\varepsilon}\right)$$

$$= \mathcal{O}\left(\sqrt{\frac{\min\{\tilde{L}, \lambda\}}{\mu}}\log\frac{1}{\varepsilon}\right)$$

as desired.

The local stochastic gradient complexity for $p = \frac{\lambda}{\lambda + \tilde{L}}$ and $\rho = \frac{1}{m}$ is

$$\mathcal{O}\left( (\rho m + (1 - \rho)) \left( \frac{1}{\rho} + \sqrt{\frac{\max\left\{ \frac{\tilde{L}}{1-p}, \frac{\lambda}{p} \right\}}{\rho \mu}} \right) \log \frac{1}{\varepsilon} \right)$$

$$= \mathcal{O}\left( \left( m + \sqrt{\frac{m(\tilde{L} + \lambda)}{\mu}} \right) \log \frac{1}{\varepsilon} \right).$$

$\square$

# F   Related work on the lower complexity bounds

**Related literature on the lower complexity bounds.**   We distinguish two main lines of work on the lower complexity bounds related to our paper besides already mentioned works [44, 21].

The first direction focuses on the classical worst-case bounds for sequential optimization developed by Nemirovsky and Yudin [35]. Their lower bound was further studied in [2, 43, 39] using information theory. The nearly-tight lower bounds for deterministic non-Euclidean smooth convex functions were obtained in [17]. A significant gap between the oracle complexities of deterministic and randomized algorithms for the finite-sum problem was shown in [51], improving upon prior works [1, 27].

The second stream of work tries to answer how much a parallelism might improve upon a given oracle. This direction was, to best of our knowledge, first explored by the work of Nemirovski [34] and gained a lots of traction decently [47, 4, 52, 12, 11] motivated by an increased interest in the applications in federated learning, local differential privacy, and adaptive data analysis.

**Remark F.1** *Concurrently with our work, a different variant of accelerated* `FedProx`—`FedSplit`— *was proposed in [40]. There are several key differences between our work: i) While Algorithm 1 is designed to tackle the problem (2),* `FedSplit` *is designed to tackle (1). ii) The paper [40] does not argue about optimality of* `FedSplit`*, while we do and iii) Iteration/communication complexity of* `FedSplit` *is* $\mathcal{O}\left( \sqrt{\frac{L}{\mu}} \log \frac{1}{\varepsilon} \right)$ *under $L$ smoothness of $f_1, \dots f_n$; such a rate can be achieved by a direct application of* `AGD`*. At the same time,* `AGD` *does not require solving the local subproblem each iteration, thus is better in this regard. However* `FedSplit` *is a local algorithm to solve (1) with the correct fixed point, unlike other popular local algorithms.*