[Reviews · NeurIPS 2020]

Review 1

Summary and Contributions: -This paper considers the problem of personalized federated learning. Specifically they consider the case of convex optimization. -They first establish lower bound results on both the communication complexity and the number local gradient oracle calls. -Next, using the lower bound results they show that an accelerated version of an algorithm similar to that of [50] has complexity equal to the lower bounds proven (up to a condition on the convexity parameters). -Since that method requires oracle access to the local proximal function - which may often be intractable - they provide a various inexact versions and prove that they also enjoys various convergence rates that are sometime tight to the lower bound. -Finally, they propose one last algorithm that combines various ideas and is optimal in the most broad range of scenarios.

Strengths: - The theoretical analysis is detailed. The inclusion of lower bounds is both helpful for establishing what is possible in general, but also in validating the quality of the various algorithms proposed in the paper. - The algorithms proposed are, in various situations, optimal (but the overlapping cases for communication vs local gradient calls are a little complicated).

Weaknesses: - It is harsh to really call this a weakness, but: by the authors own admission (line 73) the optimality results for communication and local gradient calls do not hold simultaneously. So there is a gap, and still something for future work. - The results depend on Assumption 3.1, which other related works do not require.

Correctness: So far as I can tell, full theoretical justifications are given for all results. I did not check any proofs, however, so I am accepting their veracity on faith.

Clarity: The paper is well written and logically organized. At times it became a little difficult to keep track of all the various settings and conditions and rates one observes. This is, however, something that is somewhat challenging to make simpler. A couple of typos: l.14 “Unlike a typical” —-> “unlike typical” l.146 the "." after the footnote.

Relation to Prior Work: It seems fairly clear how the work differs from prior work. They are clear in acknowledging when their methods are building on those of others. I do not have a deep knowledge of related literature, so can't personally say for sure that all relevant related work is discussed. However, the extensive discussion of related work, including an extra section in the appendix of further discussion gives me good confidence. In case it is interesting to you the “Provable Guarantees for Gradient-Based Meta-Learning” paper seems quite related to the comment about the solution of eon (2) being related to MAML. In particular they draw a connection between MAML and “meta regularization” which is of this form of a L2 parameter penalization. http://proceedings.mlr.press/v97/balcan19a/balcan19a.pdf

Reproducibility: Yes

Additional Feedback: Overall the paper is a good step in the right direction - undoubtedly this work makes progress in the understanding of personalized federated learning. I do however, have some questions. I would be interested in hearing further discussion from the authors on assumption 3.1. On a high level, what part of the logic requires this assumption to go through? The assumption seems to originate from literature outside of federated learning, is that correct? Could you discuss in what sense the assumption is "natural" for this kind of learning problem? Update: based on author feedback I am happy to confirm my overall assessment of the paper as an accept.


Review 2

Summary and Contributions: Based on the global consensus reformulation of Federated Optimization, this paper proposes lower bounds on the communication complexity and optimal algorithms for personalized Federated Learning.

Strengths: This paper provides lower complexity bounds and optimal algorithms (under specific parameter regime) for personalized FL, which further our understanding. The convergence analysis for the mentioned algorithms seems correct. Experiments are done well to illustrate some theoretical points.

Weaknesses: 1. The example derived for the lower bound and its idea behind are quite similar to that of Nesterov in [36]. And I have some questions about the example for the lower bound. In line 432, I didn’t figure out why the expression of M is matrix dependent on the parity of n. From Lemma D.1 in [19], $\nabla^{2} \psi(x)=\frac{1}{n}\left(\mathbf{I}_{n}-\frac{1}{n} e e^{\top}\right) \otimes \mathbf{I}$ is the second derivative of $\phi(x)$, which should appear in the second derivative of $M$. However, it didn’t. Is there something wrong with my understanding? What’s more, there are two identity matrices without specifying its dimension, making it hard to understand. Besides, 2. It seems that AL2SGD+ works well empirically. However, compared to its non-accelerated version L2SGD, the modification is not well clarified. By the way, it seems slightly different from L-Katyusha. 3. In the proof of Theorem 4.4 (Appendix E.2.1), to achieve the desired communication complexity and local stochastic gradient complexity, the author uses two different values of $p$ and $\rho$. My question is whether it is possible to achieve such bounds just with one choice of hyper-parameters? I think this might be very important for future research. Some typos : 1. the definition of proximal operator eq(4) seems wrong 2. line 235: “APGD2” should be the algorithm of choice for $\lambda > L = 1$ 3. line 411: the iteration complexity seems omitted. 4. line 431: it seems that in eq(9), $a$ should be $a / \lambda$. 5. line 538-539: m if $\xi=0, \xi’ =1$.

Correctness: The main issue is given in the Weaknesses part. The rest of the paper I think is good and seems correct.

Clarity: Yes

Relation to Prior Work: Yes

Reproducibility: Yes

Additional Feedback:


Review 3

Summary and Contributions: The paper studies convergence rates (upper and lower) for federated learning.

Strengths: The paper provides a comprehensive study of acceleration rates of federated proximal gradient and lower bound as well.

Weaknesses: The results are fine and are comprehensive. But my main issue with the paper is that the analysis focus is missing the point of federated learning. All the results obtained are not about how the convergence rate scales as a function of the number of participating devices/clients. If this paper were submitted in 2017 when FL just took off, I'd vote for accept. But ever since then, there has been an extensive line of work (many of them cited in the papers, so I'm not complaining about missing references) that studies convergence of FL algorithms and show the same type of rate as in a single client setting. Granted, this paper's results, to the best of my knowledge, are not obtained before. But from an optimization theoretical perspective, it is one of those results that can be obtained by plugging in the analyses for the centralized algorithm (in this case proximal gradient) and then showing that things go through in the FL setting. By itself this is fine, but it is getting a bit boring after seeing several of those papers who did the same for other FL algorithms (e.g. FedAvg, accelerated FedAvg). I feel the emphasis really need to be on how does convergence scale with the number of clients--this is the essence of the value brought forth by FL. So I'd give the rating marginally below considering the high bar of NIPS (at a lower tier conference, I would have been fine with accepting the paper).

Correctness: yes

Clarity: yes

Relation to Prior Work: yes

Reproducibility: Yes

Additional Feedback: Post rebuttal: I had read the authors' response and through the discussions with other reviewers, had decided to keep the current score.


Review 4

Summary and Contributions: This paper studies the bounds of communication and computation complexity of minimizing the mixing federated learning objective. Then the authors analyze the complexities of state-of-the-art methods under different settings and extend existing methods.

Strengths: This paper finds the lower bounds when \lambda >= 1 and analyzes the existing methods. By considering whether the lower bounds are matched, the authors propose new methods.

Weaknesses: Theorem 3.1 only considers the case when \lambda\le\frac{L}{\mu} and \lambda\ge 100\mu. The personalized federated learning mainly focuses on achieving small local losses and hence \lambda can be very small. It is inappropriate to require that \lambda\ge 100\mu. In addition, Equation (3) implies that \frac{L}{\mu}\ge 101 and hence Theorem 3.1 does not apply to well-conditioned functions. Therefore, the significance of this paper is limited. The notations are not clearly defined. For example, the "Span" in Assumption 3.1 has not been formally defined. The tick and cross symbols in Table 1 are not defined. The relationship between Table 1 and Table 2 requires further explanation. In Table 2, the authors put a tick and a cross in the same cell (the one in the 5th row and the 2nd column), which is very confusing. The proposed methods are trivially extended from existing methods.

Correctness: Yes, the theoretical results are well-expected and the experimental setting is correct.

Clarity: No. This paper is not well-organized and needs further improvement in writing.

Relation to Prior Work: Yes. The proposed methods are extended from state-of-the-art methods.

Reproducibility: Yes

Additional Feedback: Given proximal oracle or gradient oracle, the communication complexity and the computation complexity are of the same order. Does this mean that the nodes should communication after a fixed number of local iterations? What is the communication complexity when \lambda<1? The personalized federated learning mainly focuses on achieving small local losses and hence \lambda can be very small. When \lambda<1, can we simply multiply \lambda and f(x) by the same constant so that Theorem 3.1 can be applied?

[Author Response · NeurIPS 2020]

We would like to thank all reviewers for the constructive feedback. We argue that most of the issues raised are either minor or no issues at all. Before proceeding, we would like to highlight the contribution of this work that we believe was overlooked. Specifically, our work is the first to show the optimality of local methods (=most popular FL optimizers) in the non-iid regime. So far, local methods have been known to be optimal only in certain scenarios with iid data (=not a reasonable assumption for FL); thus we provide an important justification of local methods in the practical regime. See point 7) for more details. We hope that the scores will be raised accordingly.

**R1** Thanks a lot for the positive evaluation of our work!
1) Assumption 3.1. Assumption 3.1 was only used for the lower complexity bounds; it defines the algorithm class for which we provide the lower bounds. Note that virtually literally any reasonable FL optimizer satisfies the assumption: FedAvg/Local SGD, FedProx, FedSplit, SCAFFOLD, variants of (accelerated) SGD, or any algorithm that we propose or mention in our paper. Informally, we restrict ourselves to the algorithms whose iterates only lie in the span of the previously observed gradients. From the optimization perspective, it does not make sense to do not have it satisfied for any reasonable algorithm. To assume something along those lines is indeed typical in the lower-bound optimization literature. Note that a recent paper on the lower bounds for the classical FL objective assumes the same thing: `https://arxiv.org/pdf/2005.10675.pdf` (see Sec. 4.1). 2) Gap. There is still a rather small gap in a single setting. However, in most scenarios we consider, there is no gap. 3) Rest. Thanks a lot for the useful MAML reference; we will mention it! Typos: thanks, fixed!

**R2** Thank you very much for the positive evaluation of our work!
4) line 432. There are 2 typos in the appendix: a typo in the definition of $\mathbf{M}$; the term $1/2$ should be replaced with $1/n$. At the same time, there is a typo in the equation (9): there should be $\frac{n}{\lambda}$ instead of $\frac{1}{\lambda}$. Back to your question – function $\psi$ does not depend on the parity of $n$; Hessian of $\psi$ is now the first component of the equation defining $\mathbf{M}$, while functions $f_i$ depend on the parity (note that we chose them as quadratics). Thanks a lot for noticing this! We will also add a subscript to highlight the dimensions of identity matrices. 5) AL2SGD+. The rationale behind AL2SGD+ is: take the variance reduced gradient estimator from L2SGD+, and feed it into L-Katyusha. AL2SGD+ looks different than L-Katyusha is since we wrote it from the FL perspective. 6) Different values of $p, \rho$. Setting those parameters to be identical yields a suboptimal complexity in certain scenarios, this is why we chose them differently. Rest: fixed, thanks!

**R3** A few concerns were raised about the contribution of this work. Below we argue that those are not a concern.
7) Not focusing on the "scale with the number of clients". Our paper provides a different result that is much more valuable to the FL community! In particular, we are the first to show that local methods (variants of FedAvg/local SGD or FedProx) are optimal in the heterogeneous (non-iid) data regime both in terms of the communication and local computation complexity. We believe this is of great import in the FL area as i) local methods are known to be suboptimal in the non-iid regime for the classical non-personalized FL (see Woodworth et al. 2020 `https://arxiv.org/pdf/2006.04735.pdf` for example) and ii) local methods are the most popular FL optimizers in practice. By showing the optimality of local methods, we justify the common FL practices (i.e., use of local methods on non-iid data), which we believe is of greater value than the focus on the number of clients. 8) "Results that can be obtained by plugging in the analyses for the centralized algorithm and then showing that things go through in the FL setting". The lower bound we propose can not be obtained by any means from the known results. The key thing is that both upper and lower bounds are new (the upper bound was somewhat easier to get), and they are matching in most cases (see the reply to R1). While it is true that complexities of some methods are not hard to get (which we clearly state in the paper), it was non-trivial to find out the algorithms that match the proposed lower bounds, as well as to set them up properly so that the bound is achieved.

**R4** The reviewer raised several concerns. Below, we argue that neither of those are major.
9) Thm 3.1. We missed a comma in the theorem statement: we meant $L \geq \mu$, $\lambda \geq \mu$ instead of the original $L \geq \mu\lambda \geq \mu$. Sorry! Those are the only constraints we have. As you noticed, there are some extra constants (100 and 101) over there for the bound to be meaningful. First of all, our results hold for a finite-dimensional problem; those constants can be improved by blowing up the dimension to infinity, which we chose to avoid. Secondly, even for the infinite-dimensional construction, many distributed lower bounds incur a similar (but smaller) constant (see, for example [20]). Next, requiring $\lambda \geq 100\mu$ is usually fine since the strong convexity constant $\mu$ is often tiny (from a learning theory perspective, one should the weight for the Tikhonov's regularization is inversely proportional to the dataset size). Lastly, we shall stress that this (rather mild) parameter restriction only holds for the lower complexity bounds. 10) Notations and tables. We will clarify those, thanks! 11) The proposed methods are trivially extended from existing methods. See the point 8). 12) Additional feedback, Q1. Indeed, it means that the optimal algorithm requires only a constant (=small) number of local oracle queries in between the communications. Note that the computation and communication complexities are not the same for the local gradient oracle once $\lambda < L$; in such a case, one needs to perform more local gradient steps in between of the communications. 13) Additional feedback, Q2. We never assume that $\lambda \geq 1$. We only assume that $\lambda \geq \mu$ for the lower bound; the algorithm complexities hold regardless of the choice of $\lambda$.

[Meta-Review · NeurIPS 2020]

The submission studies lower bounds on convergence rates for federated learning, including for accelerated proximal gradient. The lower bounds established consider the number of oracle calls and the communication complexity. Also it is proven that there exists a method matching the lower bound. The rates are new and the paper is well-written. Interesting avenues for further research are a dependency on the number of clients. For a final version of the paper, besides the clarifications pointed out by reviewers discussing the range of parameters $\lambda$ and $\mu$ needs to be addressed, a discussion on how the number of clients affects convergence (i.e. in which constants the of the complexity estimates the nr. of clients has an influence) would further improve the paper.